# Phosphoregulation of DSB-1 mediates control of meiotic double-strand break activity

Heyun Guo[1], Ericca L Stamper[2,3,4,5†], Aya Sato-Carlton[1], Masa A Shimazoe[1,6‡§], Xuan Li[1], Liangyu Zhang[2,3,4,5], Lewis Stevens[7], KC Jacky Tam[1#], Abby F Dernburg[2,3,4,5], Peter M Carlton[1,8,9]*

[1]Graduate School of Biostudies, Kyoto University, Yoshidakonoe, Sakyo, Kyoto, Japan; [2]Department of Molecular and Cell Biology, University of California, Berkeley, United States; [3]Howard Hughes Medical Institute, Chevy Chase, United States; [4]California Institute for Quantitative Biosciences, Berkeley, United States; [5]Division of Biological Systems and Engineering, Lawrence Berkeley National Laboratory, Berkeley, United States; [6]Department of Science, Kyoto University, Kyoto, Japan; [7]Institute of Evolutionary Biology, Ashworth Laboratories, School of Biological Sciences, University of Edinburgh, Edinburgh, United Kingdom; [8]Radiation Biology Center, Kyoto University, Kyoto, Japan; [9]Institute for Integrated Cell-Material Sciences (iCeMS), Kyoto University, Kyoto, Japan

**\*For correspondence:**
carlton.petermark.3v@kyoto-u.ac.jp

**Present address:** [†]Harriet L. Wilkes Honors College, Florida Atlantic University, Jupiter, United States; [‡]Genome Dynamics Laboratory, National Institute of Genetics, Mishima, Japan; [§]Department of Genetics, School of Life Science, SOKENDAI, Mishima, Japan; [#]Centre for Regenerative Medicine, University of Edinburgh, Edinburgh, United Kingdom

**Abstract** In the first meiotic cell division, proper segregation of chromosomes in most organisms depends on chiasmata, exchanges of continuity between homologous chromosomes that originate from the repair of programmed double-strand breaks (DSBs) catalyzed by the Spo11 endonuclease. Since DSBs can lead to irreparable damage in germ cells, while chromosomes lacking DSBs also lack chiasmata, the number of DSBs must be carefully regulated to be neither too high nor too low. Here, we show that in *Caenorhabditis elegans*, meiotic DSB levels are controlled by the phosphoregulation of DSB-1, a homolog of the yeast Spo11 cofactor Rec114, by the opposing activities of PP4$^{PPH-4.1}$ phosphatase and ATR$^{ATL-1}$ kinase. Increased DSB-1 phosphorylation in *pph-4.1* mutants correlates with reduction in DSB formation, while prevention of DSB-1 phosphorylation drastically increases the number of meiotic DSBs both in *pph-4.1* mutants and in the wild-type background. *C. elegans* and its close relatives also possess a diverged paralog of DSB-1, called DSB-2, and loss of *dsb-2* is known to reduce DSB formation in oocytes with increasing age. We show that the proportion of the phosphorylated, and thus inactivated, form of DSB-1 increases with age and upon loss of DSB-2, while non-phosphorylatable DSB-1 rescues the age-dependent decrease in DSBs in *dsb-2* mutants. These results suggest that DSB-2 evolved in part to compensate for the inactivation of DSB-1 through phosphorylation, to maintain levels of DSBs in older animals. Our work shows that PP4$^{PPH-4.1}$, ATR$^{ATL-1}$, and DSB-2 act in concert with DSB-1 to promote optimal DSB levels throughout the reproductive lifespan.

## Editor's evaluation

The connection between double-strand break (DSB) formation and chromosome pairing/synapsis during meiosis is not fully understood. In this paper, the authors show that the formation of DSBs is regulated by the DNA damage response (DDR) machinery. The paper will be of interest to the broad meiosis and DDR communities.

## Introduction

To reduce chromosome number from diploid to haploid during sexual reproduction, homologous chromosomes must segregate to different daughter cells in the first division of meiosis. Most organisms achieve this segregation by linking homologous chromosomes with chiasmata, exchanges of continuity between chromatids that derive from repair of programmed double-strand breaks (DSBs). DSBs are created by the conserved endonuclease Spo11 acting in concert with an array of cofactors (*Dernburg et al., 1998*; *Keeney et al., 1997*; *Panizza et al., 2011*; *Yadav and Claeys Bouuaert, 2021*). The initiation of DSBs needs to be strictly controlled, due to their deleterious potential: not only can unrepaired breaks lead to apoptosis (*Bhalla and Dernburg, 2005*; *Roeder and Bailis, 2000*), but unfavorable repair mechanisms such as non-homologous end-joining or non-allelic homologous recombination acting on DSBs (*Kim et al., 2016*) can lead to genome rearrangement or deletions. Despite these dangers, however, every chromosome pair requires at least one crossover for proper segregation, so DSB initiation must be allowed to occur until this condition has been met. Accordingly, DSBs must be regulated in space and time to achieve a number that is not too high, but not too low. How this regulation occurs, that is how each species enforces the correct level of DSBs they require (*Kauppi et al., 2013*) remains an unsolved mystery.

A large body of work has shown that the DNA damage sensor kinases ATM and/or ATR control DSB initiation and repair at multiple levels in mammals (*Lange et al., 2011*), *Drosophila* (*Joyce et al., 2011*), budding yeast (*Carballo et al., 2013*; *Garcia et al., 2015*; *Zhang et al., 2011*), and other organisms. When a break occurs, ATM/ATR (yeast Tel1/Mec1) locally phosphorylate many substrates leading to a local reduction in further DSB formation (DSB interference) in budding yeast (*Garcia et al., 2015*; *Mohibullah and Keeney, 2016*). ATR(Mec1) activity induced by replication stress has also been shown to reduce the chromosome loading of Rec114, a Spo11 accessory factor, in budding yeast (*Blitzblau and Hochwagen, 2013*). In mice, both ATM and ATR act to remove recombination factors from the vicinity of DNA breaks and suppress DSB initiation (*Dereli et al., 2021*; *Lange et al., 2011*). However, in both mice and budding yeast it is unknown whether any phosphatase counteracts or regulates the anti-DSB activity of these kinases.

Rec114, originally discovered through screens in budding yeast to identify genes required for initiation of meiotic recombination (*Malone et al., 1991*; *Menees and Roeder, 1989*), acts in concert with Mei4 and Mer2, together referred to as the RMM complex, to promote DSB initiation (*Kumar et al., 2015*; *Li et al., 2006*). Homologs of yeast Rec114 include mouse Rec114 (*Kumar et al., 2018*; *Kumar et al., 2015*), fission yeast Rec7 (*Molnar et al., 2001*), and *Caenorhabditis elegans* DSB-1 and DSB-2 (*Rosu et al., 2013*; *Stamper et al., 2013*; *Tessé et al., 2017*), all of which are required for meiotic DSB formation. A homolog of Mei4, DSB-3, is also required for DSB formation in *C. elegans* (*Hinman et al., 2021*), but no nematode homolog of Mer2 has been identified as of this writing. While the exact mechanism of DSB promotion by the RMM complex remains obscure, recent evidence suggests it may act as a scaffold for the Spo11 core complex (*Claeys Bouuaert et al., 2021*; *Johnson et al., 2021*).

In budding yeast and many other organisms, mutations that abolish DSB formation or processing also block synapsis, the polymerization of a protein macroassembly called the synaptonemal complex (SC) that holds chromosomes together in meiosis. This dependence has led to the conclusion that synapsis between homologous chromosomes is dependent on successful meiotic recombination in these organisms (*Baudat et al., 2000*; *Kleckner, 1996*; *Roeder, 1995*; *Romanienko and Camerini-Otero, 2000*; *Tessé et al., 2003*). In contrast, in *C. elegans* and *Drosophila melanogaster*, homologous chromosomes can pair and synapse in the complete absence of recombination (*Dernburg et al., 1998*; *McKim et al., 1998*), and thus it has been suggested that these organisms achieve recombination-independent pairing and synapsis. However, while *C. elegans* can achieve homologous synapsis in the absence of DSB formation, recent evidence suggests that this is equivalent to an early form of dynamic and unstable synapsis, which is normally later stabilized by DSB-induced recombination (*Machovina et al., 2016*; *Pattabiraman et al., 2017*; *Roelens et al., 2015*). The extent to which recombination contributes to homologous pairing and synapsis in *C. elegans* is not well understood.

Protein substrates phosphorylated by ATM and ATR are known to be dephosphorylated in many contexts by the highly conserved serine/threonine protein phosphatase 4 (PP4) (*Hustedt et al., 2015*; *Keogh et al., 2006*; *Kim et al., 2011*; *Lee et al., 2010*). Previously, we have shown that PP4 in *C. elegans* (PPH-4.1) is required for viability-supporting levels of DSB initiation as well as homologous pairing and synapsis during meiotic prophase (*Sato-Carlton et al., 2014*). PP4 has also been shown

to regulate diverse meiotic events in other organisms, coordinating loss of centromere pairing with recombination in budding yeast (*Falk et al., 2010*) and suppressing crossover formation in *Arabidopsis* (*Nageswaran et al., 2021*).

In this work, we show that the DSB-promoting activity of DSB-1 is controlled by both PPH-4.1 and ATR (*C. elegans* ATL-1): meiotic DSB levels are decreased by the phosphorylation of DSB-1, but drastically increase when DSB-1 cannot be phosphorylated. During meiotic prophase, DSB-1 is phosphorylated in an ATL-1-dependent manner to inhibit DSB formation and protect the genome against excessive DSBs. In contrast, DSB-1 is dephosphorylated in a PPH-4.1-dependent manner, thereby promoting a number of DSBs sufficient to form a crossover on each chromosome pair and allow proper chromosome segregation. Since ATM/ATR kinases are known to be activated by DNA breaks, our model predicts that activated ATR kinase turns off the DSB machinery via DSB-1 phosphorylation once sufficient levels of recombination intermediates are generated. This feedback mechanism could tune DSB levels to ensure the formation of crossovers via phosphoregulation of DSB-1. Moreover, we find that the homologous pairing and synapsis defects in *pph-4.1* mutants are significantly rescued when DSB levels are increased by a non-phosphorylatable allele of *dsb-1*, adding to the growing evidence that DSBs can strengthen homologous synapsis in *C. elegans*. Our results shed light on fundamental mechanisms of meiotic chromosome dynamics regulated by contrasting kinase and phosphatase activities.

## Results

### DSB-1 undergoes phosphorylation which is prevented by PPH-4.1[PP4] phosphatase

We previously showed that the activity of the PP4 phosphatase catalytic subunit, PPH-4.1, is necessary for normal levels of DSB initiation in *C. elegans* (*Sato-Carlton et al., 2014*). This result suggested that a hyperphosphorylated substrate may inhibit DSBs in *pph-4.1* mutants. Previous work in budding yeast showed that the Spo11 cofactor Rec114 is phosphorylated by ATM[Tel1] and ATR[Mec1] (*Carballo et al., 2013*) in response to meiotic DSBs. In *C. elegans*, two recently diverged orthologs of Rec114, DSB-1 and DSB-2, are also required for normal DSB formation (*Rosu et al., 2013*; *Stamper et al., 2013*); DSB-1 is absolutely required for DSBs, whereas loss of DSB-2 still allows a low level of DSB initiation. To examine if PPH-4.1 regulates DSB levels through DSB-1, we determined whether loss of *pph-4.1* leads to DSB-1 hyperphosphorylation. We performed western blotting on extracts from animals carrying a GFP fusion of DSB-1 at the endogenous locus, comparing animals treated with RNAi against *pph-4.1* to control animals treated with an empty RNAi vector. To ensure sufficient knockdown against *pph-4.1*, RNAi is started in the P0 generation on synchronized L4 larvae, and continued until extracts were made from adults of the next (F1) generation. The effectiveness of *pph-4* RNAi was verified by observing univalents in diakinesis oocytes (*Figure 1—figure supplement 1A*). Two major bands were apparent in both treatments: one at the predicted size for DSB-1, and one more slowly migrating (*Figure 1A*, left). In extracts from the *pph-4.1* RNAi-treated animals, the proportion of the upper band was significantly stronger than the band at the expected size (*Figure 1A*, right). This result suggests that depletion of PPH-4.1 leads to hyperphosphorylation of DSB-1. To test whether our observations were influenced by the fusion of GFP to DSB-1 protein, we next performed western blots using polyclonal antibodies directed against DSB-1 itself. Since *pph-4.1* mutants are mostly embryonic inviable, in order to obtain sufficient *pph-4.1*-deficient material, we made lysates from a mixed population of *pph-4.1* homozygous and balanced heterozygous mutant adults further treated with *pph-4.1* RNAi, and obtained a similar shift compared to a wild-type control (*Figure 1—figure supplement 1B*). To test whether the slow-migrating band was caused by protein phosphorylation, we added $\lambda$-phosphatase to lysates from *gfp-dsb-1* worms treated with *pph-4* RNAi. Phosphatase treatment abolished the slow-migrating band, showing that it was a phosphorylated fraction of DSB-1 (*Figure 1B*). Taken together, these results strongly suggest that phosphorylated DSB-1 protein is normally dephosphorylated in a PPH-4.1-dependent manner in wild-type animals.

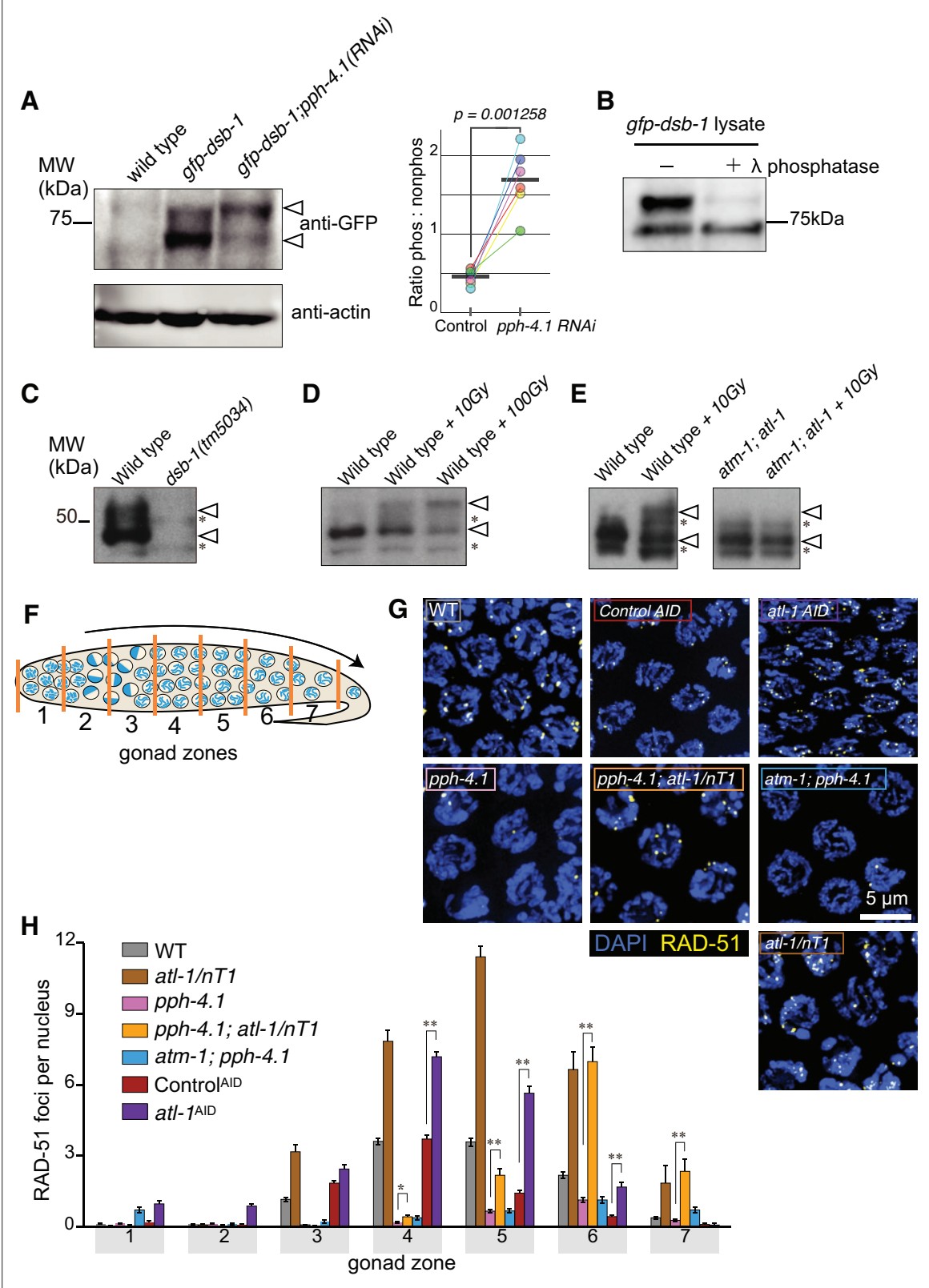

**Figure 1.** DSB-1 is phosphoregulated in a PPH-4.1$^{PP4}$- and ATL-1$^{ATR}$-dependent manner and ATL-1$^{ATR}$ kinase antagonizes PPH-4.1$^{PP4}$ phosphatase. (**A**) *Left:* Western blot of GFP-fused DSB-1 probed with α-GFP. GFP-DSB-1 detected in extracts from wild-type and *gfp-dsb-1* worms (24 hr post-L4 stage) with either control RNAi or *pph-4.1* RNAi treatment. A total protein amount of 97 µg was loaded in each lane. Arrowheads indicate two specific bands in the blot. Loading controls (α-actin) are shown at bottom. *Right:* Quantified band intensity ratio of phos-GFP-DSB-1 to non-phos-GFP-DSB-1

*Figure 1 continued on next page*

*Figure 1 continued*

in each genotype. Mean (bar) and data points are from six biological replicates. Significance was assessed via two-tailed t test. (*Figure 1—source data 1*, *Figure 1—source data 2*). (**B**) Blot of GFP-DSB-1 worm lysate from young (1 day post-L4 stage) adult *pph-4.1* RNAi-treated worms with or without $\lambda$-phosphatase treatment. (*Figure 1—source data 3*). (**C, D, E**) Western blots of endogenous DSB-1 probed with α-DSB-1 antibodies in wild type (**N2**), *dsb-1(tm5034), atm-1(gk186), atl-1(tm853)*, or *atm-1(gk186); atl-1(tm853)* with combination of γ-irradiation (10 Gy in panel E, and 10 or 100 Gy in panel D as indicated). Lysate of 50 worms at 24 hr post-L4 stage was loaded in each lane. Asterisks show non-specific bands, and arrowheads indicate two specific bands. (*Figure 1—source data 4*, *Figure 1—source data 5*, *Figure 1—source data 6*). (**F**) Schematic showing a hermaphrodite gonad divided into seven equally sized zones for RAD-51 focus scoring. (**G**) Immunofluorescence images of RAD-51 foci in mid-pachytene nuclei (zone 5) of the indicated genotypes. Scale bar, 5 µm. (**H**) Quantification of RAD-51 foci in the germlines of the genotypes indicated in (**G**). Data are presented as mean ± SEM from at least three biological replicates. Seven gonads were scored for wild type and *pph-4.1(tm1598)*, three gonads were scored for *atl-1(tm853)/nT1, atl-1^{AID}, control^{AID}* as well as *pph-4.1(tm1598); atl-1(tm853)/nT1* double mutants, and four gonads were scored in *atm-1(gk186); pph-4.1(tm1598)*. The numbers of nuclei scored in zones 1–7 were as follows: for wild type, 420, 453, 377, 375, 345, 271, 296; for *pph-4.1(tm1598)*, 433, 423, 422, 413, 355, 322, 208; for *atl-1(tm853)/nT1*, 103, 137, 145, 115, 97, 75, 40; for *atl-1^{AID}*, 161, 193, 180, 241, 204, 117, 37; for *control^{AID}*, 143, 188, 233, 223, 192, 109, 28; for *pph-4.1(tm1598); atl-1(tm853)/nT1*, 126, 121, 98, 100, 94, 86, 49; for *atm-1(gk186);pph-4.1(tm1598)*, 123, 153, 167, 140, 161, 156, 123. Significance was assessed via two-tailed t test, **p<0.01, ****p<0.0001 (*Figure 1—source data 7*).

The online version of this article includes the following source data and figure supplement(s) for figure 1:

**Source data 1.** Intensity of phos-GFP-DSB-1 and non-phos-GFP band in *Figure 1A*.

**Source data 2.** Western blotting raw images in *Figure 1A*.

**Source data 3.** Western blotting raw images in *Figure 1B*.

**Source data 4.** Western blotting raw images in *Figure 1C*.

**Source data 5.** Western blotting raw images in *Figure 1D*.

**Source data 6.** Western blotting raw images in *Figure 1E*.

**Source data 7.** RAD-51 foci numbers graphed in *Figure 1H*.

**Figure supplement 1.** Validation of effectiveness of *pph-4* RNAi and western blots of DSB-1.

**Figure supplement 1—source data 1.** Western blotting raw images in *Figure 1—figure supplement 1B*.

**Figure supplement 1—source data 2.** Western blotting raw images in *Figure 1—figure supplement 1C*.

**Figure supplement 1—source data 3.** Western blotting raw images in *Figure 1—figure supplement 1D*.

**Figure supplement 2.** Double-strand break (DSB) formation in *atl-1* and *atm-1* mutants.

**Figure supplement 2—source data 1.** RAD-51 foci numbers graphed in *Figure 1—figure supplement 2C*.

## ATL-1^{ATR} kinase opposes the DSB initiation activity of PPH-4.1^{PP4} phosphatase

To examine if ATM/ATR kinase may phosphorylate DSB-1 and thus antagonize PPH-4.1 phosphatase activity, we performed western blots to detect endogenous DSB-1 in *atm-1^{ATM}; atl-1^{ATR}* double mutants in combination with γ-ray irradiation to create DNA DSBs, thereby activating ATM/ATR kinases. A phospho-DSB-1 band visible in wild-type animals showed a relatively increased fraction in a manner dependent on γ-ray dose (*Figure 1C and D*). While 10 Gy of γ-rays sufficed to show a slow-migrating band in wild-type lysates, this band was not detected in *atm-1; atl-1* double mutants at the same dose of γ-irradiation, suggesting that DSB-1 becomes phosphorylated in an ATM/ATR-dependent manner (*Figure 1E*). We also conducted the same γ-ray irradiation in mutants that do not make DSBs (*spo-11(me44), htp-3(tm3655), chk-2(me64), rad-50(ok197), mre-11(ok179)*) (*Chin and Villeneuve, 2001*; *Dernburg et al., 1998*; *Goodyer et al., 2008*; *Hayashi et al., 2007*; *MacQueen and Villeneuve, 2001*). In all five of these mutant backgrounds, the phosphorylated band was absent before irradiation. Upon γ-irradiation, a phosphorylated band in *spo-11, htp-3*, and *chk-2* mutants became visible, but was still undetectable in *mre-11* mutants and extremely faint in *rad-50* mutants (*Figure 1—figure supplement 1C*, D). In other words, in *mre-11* and *rad-50* mutants, DSB-1 was not phosphorylated, or was phosphorylated very weakly, even in the presence of DSBs. This is consistent with the previous observation that the MRN (Mre11/Rad50/Nbs1) complex plays an important role in activating ATM/ATR kinases (*Duursma et al., 2013*; *Garcia-Muse and Boulton, 2005*; *Lee and Paull, 2004*; *Uziel et al., 2003*). These data further reinforce the hypothesis that DSB-1 phosphorylation depends on DSBs and activation of ATM/ATR kinases.

To further examine if ATM/ATR kinases antagonize PPH-4.1 phosphatase to regulate DSB-1 activity, we combined the *pph-4.1(tm1598)* mutation with the *atm-1* or *atl-1* mutations. We assessed DSB

formation in these double mutants by performing immunofluorescence against the strand-exchange protein RAD-51. Since the *C. elegans* gonad contains nuclei from all stages of meiotic prophase arranged sequentially along the distal-proximal axis, we divided the gonad into seven zones of equal size (*Figure 1F*) and counted the number of RAD-51 signals in nuclei within each zone to assess the kinetics of DSB initiation. As shown previously, mutation of *pph-4.1* led to a drastic decrease of the number of foci compared to wild-type controls (*Figure 1G and H*). Homozygous null mutation of ATM (*atm-1* in *C. elegans*) in a *pph-4.1* background only marginally increased the number of DSBs in very late pachytene. Since homozygous mutation of ATR (*atl-1* in *C. elegans*) leads to severe mitotic defects and aneuploidy in the germline due to replication errors (*Garcia-Muse and Boulton, 2005*; *Figure 1—figure supplement 2A*), we used a heterozygous mutation to bypass this effect. We found that heterozygous mutation of *atl-1* in a *pph-4.1* mutant led to a strong recovery of RAD-51 foci (*Figure 1G and H*). These results suggest that ATR kinase normally acts to suppress formation of meiotic DSBs, and this activity is opposed by PPH-4.1. Consistent with this hypothesis, we found that auxin-induced depletion (*Zhang et al., 2015*) of AID-tagged ATL-1 led to a significant increase in RAD-51 foci in mid-prophase compared to auxin-treated controls (*Figure 1H*). Similarly, we found that a heterozygous null *atl-1(tm853)* mutation led to a significant increase in overall DSB number in the presence of wild-type *pph-4.1* (*Figure 1H*). The extra DSBs seen upon loss of ATL-1 are not a result of mitotic DNA damage, since the premeiotic zone in both auxin-depleted and *atl-1* heterozygous germlines is mostly free of RAD-51 foci (*Figure 1H*, *Figure 1—figure supplement 2B*), as is also the case in the wild type. In *C. elegans*, *atm-1* homozygous null animals derived from heterozygous mothers are superficially wild type and fertile. However, *atm-1* mutants are sensitive to DNA damage-inducing reagents, and when maintained homozygously for more than 20 generations, they develop genomic instability and embryonic inviability (*Jones et al., 2012*). To assess ATM-1's contribution to DSB formation during meiotic prophase, we examined the null allele *atm-1(gk186)* in a *rad-54(ok615)* mutant background, in which DSBs are initiated but RAD-51 cannot be removed from recombination intermediates (*Mets and Meyer, 2009*; *Miyazaki et al., 2004*). We found that *atm-1; rad-54* germlines showed a significant delay in RAD-51 loading in early pachytene compared to *rad-54* single mutants but eventually showed a level of foci slightly exceeding that of the control in late pachytene (*Figure 1—figure supplement 2C*). Initial delay in RAD-51 foci appearance is consistent with a previously described role of ATM-1 in timely loading of RAD-51 (*Li and Yanowitz, 2019*). Taken together, these results show that ATL-1 plays the major role in antagonizing the DSB-promoting function of PPH-4.1, while the role of ATM-1 is less significant.

## DSB-1 possesses conserved SQ motifs, and its non-phosphorylatable mutants rescue the phenotypes of PP4 mutants

DSB-1 possesses five ATM/ATR consensus motifs ([ST]Q), two of which are highly positionally conserved in the genus *Caenorhabditis* (*Figure 2A*, *Figure 2—figure supplement 1*). These SQ sites are dispersed within a large region predicted to be intrinsically disordered (*Figure 2A*). Moreover, DSB-1 contains five copies of the FXXP motif, a conserved docking site for PP4 (*Karman et al., 2020*; *Ueki et al., 2019*); all DSB-1 orthologs shown also contain at least one copy of this motif in their disordered region (*Figure 2—figure supplement 1*). The presence of these consensus motifs raises the possibility that PPH-4.1 may dephosphorylate DSB-1 at one or more [ST]Q sites to increase DSB-promoting activity. To examine whether hyperphosphorylation of DSB-1 is responsible for loss of DSB initiation in *pph-4.1* mutants, we constructed a nonphosphorylatable mutant allele, *dsb-1(5A)*, in which all five SQ serines were replaced by alanine (*Figure 2A*). This non-phosphorylatable allele was fully viable when homozygous, demonstrating that these substitutions do not compromise the activity of DSB-1 protein (*Table 1*). We then examined RAD-51 foci in *dsb-1(5A); pph-4.1* animals to assess DSB formation. Germlines homozygous for both *pph-4.1* and *dsb-1(5A)* showed a drastically higher number of RAD-51 foci compared to *pph-4.1* single mutants (*Figure 2B and C*). Indeed, single mutants of *dsb-1(5A)* alone showed significantly elevated DSB levels compared to wild-type controls, suggesting that phosphorylation of these serine residues acts in wild-type germlines to limit the number of DSB initiations. We noted that the appearance of RAD-51 foci peaks is delayed in *pph-4.1* and *dsb-1(5A); pph-4.1*, as well as in *pph-4.1; atl-1/nT1* (*Figure 1H*, *Figure 2C*). Previous studies have shown that PP4 homologs are involved in resection of DSBs and loading of RAD-51 in yeast and mammals during the mitotic cell cycle (*Kim et al., 2011*; *Lee et al., 2010*; *Villoria et al., 2019*). DSB repair is significantly

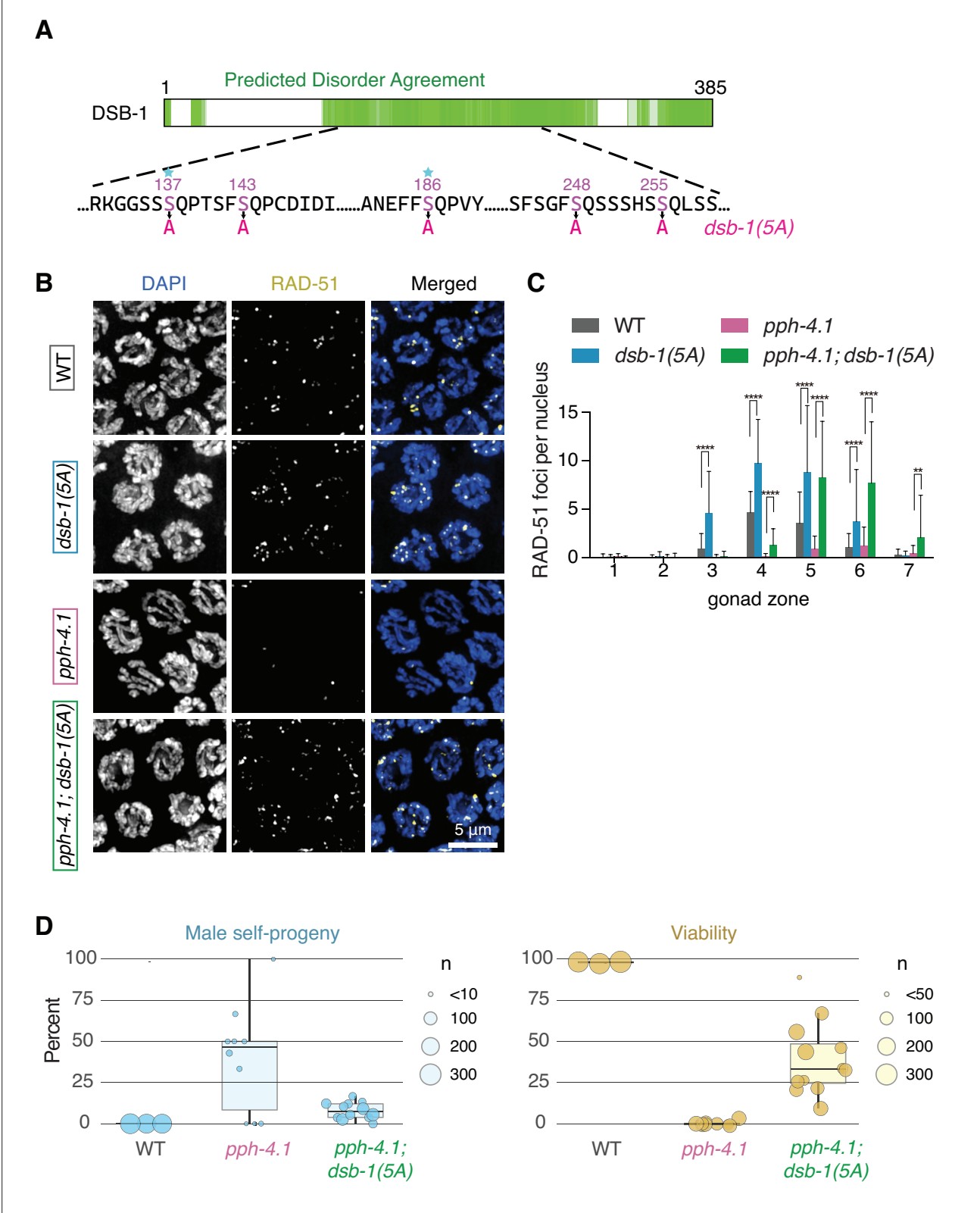

**Figure 2.** The *dsb-1(5A)* mutation rescues double-strand break (DSB) defect and viability loss of *pph-4.1* mutants. (**A**) A schematic diagram of the DSB-1 protein sequence. Green regions indicate intrinsically disordered regions from the D²P² database (*Oates et al., 2013*). Five serines which were mutated into alanines in *dsb-1(5A)* within the SQ sites are shown in magenta, and sites conserved in 10 or more of the 11 *Caenorhabditis* in the *elegans* group (see **Figure 2—figure supplement 1**) are indicated with a star. (**B**) Immunofluorescence images of RAD-51 foci in mid-pachytene nuclei of indicated

*Figure 2 continued on next page*

*Figure 2 continued*

genotypes. Scale bar, 5 µm. (**C**) Quantification of RAD-51 foci in the gonads of indicated genotypes in (**B**). Data are presented as mean ± SEM; four gonads were scored in each genotype; the numbers of nuclei scored in zones 1–7 were as follows: for wild type, 162, 201, 204, 230, 209, 155, 105; for *pph-4.1(tm1598)*, 145, 174, 201, 188, 167, 149, 74; for *dsb-1(5A)*, 175, 220, 197, 163, 143, 116, 90; for *pph-4.1(tm1598); dsb-1(5A)*, 171, 152, 121, 180, 185, 137, 85. Significance were assessed via the two-tailed t tests, **p<0.01, ****p<0.0001 (*Figure 2—source data 1*). (**D**) *Left:* Male progeny percentage indicating the rate of X chromosome nondisjunction during meiosis in wild type, *pph-4.1(tm1598)* and *pph-4.1(tm1598); dsb-1(5A)* mutants; circle size corresponds to total number of adult animals scored. *Right:* Embryonic viability percentage of the indicated genotypes; circle size corresponds to total number of eggs laid. The center indicates the median, the box denotes the 1st and 3rd quartiles, and the vertical line shows the 95% confidence interval (*Figure 2—source data 2*).

The online version of this article includes the following source data and figure supplement(s) for figure 2:

**Source data 1.** RAD-51 foci numbers graphed in *Figure 2C*.

**Source data 2.** Brood size and number of viable progeny in the genotypes indicated in *Figure 2D*.

**Figure supplement 1.** Sequence alignment of DSB-1 orthologs.

**Figure supplement 2.** Double-strand break (DSB) formation in *dsb-2* non-phosphorylatable mutant.

---

delayed in budding yeast PP4 mutants during meiotic prophase (*Falk et al., 2010*). We have previously observed meiotic cell cycle delays in *pph-4.1* mutants at the 24 hr post-L4 stage (*Sato-Carlton et al., 2014*). Since PP4 is known to have diverse substrates, PPH-4.1 may also contribute to timely processing of recombination intermediates in meiosis and/or may regulate cell cycle progression, leading to a delay in RAD-51 loading in all strains lacking PPH-4.1.

We next quantitatively scored the meiotic competence and embryonic viability of *dsb-1(5A); pph-4.1* double mutants. Since meiotic nondisjunction of the X chromosome leads to production of males (with a single X chromosome) among the self-progeny of *C. elegans* hermaphrodites, we examined the frequency of male progeny in *pph-4.1; dsb-1(5A)* mutants, and found that it decreased significantly compared to *pph-4.1* single mutants (*Figure 2D*, left). While *pph-4.1* single mutants have a very low embryonic viability of 2% on average, the viability of *pph-4.1; dsb-1(5A)* double mutants increased to 41% (*Figure 2D*, right; *Table 1*). Taken together, our results strongly suggest that DSB-1 is dephosphorylated in a PPH-4.1-dependent manner to promote DSB formation.

To test whether constitutively phosphorylated DSB-1 results in a decreased number of DSBs, we constructed *dsb-1* phosphomimetic mutants by substituting the serines of the five SQ sites with either aspartic acid or glutamic acid. However, both *dsb-1* phosphomimetic mutants exhibit wild-type levels of both RAD-51 foci and embryonic viability (data not shown). It is therefore likely that these phosphomimetic mutations do not functionally simulate the phosphorylation of DSB-1.

DSB-2, the paralog of DSB-1, also possesses four potential ATM/ATR phosphorylation sites (SQs) in its predicted disordered region. In contrast to *dsb-1(5A)* mutants, *dsb-2* non-phosphorylatable mutants, (*dsb-2(S110A_S116A_S143A_S167A)*,

**Table 1.** Embryonic viability, male progeny percentage indicating the rate of X chromosome nondisjunction, and total number of scored embryos is shown for hermaphrodite self-progeny of the indicated genotypes (*Table 1—source data 1*).

| Genotype | Embryonic viability (%) | Male percentage (%) | Total # eggs scored |
|---|---|---|---|
| WT | 99.28 | 0.04 | 1990 |
| gfp-dsb-1 | 99.10 | 0.19 | 2578 |
| dsb-1(1A) | 98.47 | 0.15 | 3427 |
| dsb-1(2A) | 98.25 | 0.00 | 1347 |
| dsb-1(3A) | 99.29 | 0.19 | 2056 |
| dsb-1(5A) | 99.22 | 0.00 | 2546 |
| pph-4.1(tm1598) | 2.00 | 45.60 | 1424 |
| dsb-2(me96) | 39.55 | 13.85 | 3413 |
| dsb-2(me96); dsb-1(1A) | 93.96 | 1.40 | 3370 |
| dsb-2(me96); dsb-1(2A) | 83.12 | 3.31 | 2413 |
| dsb-2(me96); dsb-1(3A) | 97.56 | 0.64 | 2395 |
| dsb-2(me96); dsb-1(5A) | 98.64 | 0.04 | 1596 |
| pph-4.1(tm1598); dsb-1(1A) | 7.39 | 32.21 | 496 |
| pph-4.1(tm1598); dsb-1(5A) | 40.89 | 7.89 | 1273 |

The online version of this article includes the following source data for table 1:

**Source data 1.** Brood size and number of viable progeny of the genotypes indicated in *Table 1*.

hereafter called *dsb-2(4A)*) in which all four SQ serines were replaced by alanine, did not show increased RAD-51 foci compared to the wild type, suggesting that DSB-2 is refractory to phosphoregulation (*Figure 2—figure supplement 2*).

## DSB-1 non-phosphorylatable mutants rescue the homologous pairing and synapsis defect of PP4 mutants

The striking increase in viability we observed in *pph-4.1; dsb-1(5A)* animals was somewhat surprising, given our previous finding that autosomal pairing is severely reduced in *pph-4.1* single mutants (*Sato-Carlton et al., 2014*). Pairing of an autosome (chromosome V) assessed by fluorescence in situ hybridization (FISH) did not exceed 25% in *pph-4.1* mutants, giving an expected probability of less than 0.1% for all five autosomes to pair homologously, assuming similar rates of pairing for the other autosomes. Actual measurements of synapsis and bivalent numbers at diakinesis in *pph-4.1* mutants support this expectation, since formation of six bivalents as in wild-type worms is extremely rarely seen (*Sato-Carlton et al., 2014*). We therefore hypothesized that the elevated DSB levels in *pph-4.1; dsb-1(5A)* mutants might promote bivalent formation by increasing the level of homologous pairing. To test this hypothesis, we carried out FISH against the 5S rDNA locus on chromosome V to assess the progression of homologous pairing over time, in *pph-4.1* compared to *pph-4.1; dsb-1(5A)* mutants. While the *pph-4.1* mutant showed very low homologous pairing as previously shown (*Sato-Carlton et al., 2014*), the *pph-4.1; dsb-1(5A)* double mutant rescued pairing to a significant degree, up to 70%, and the *dsb-1(5A)* single mutant showed wild-type timing and levels of homologous pairing (*Figure 3A and B*). By immunostaining against the protein ZIM-3, which marks the pairing center end of chromosomes I and IV (*Phillips and Dernburg, 2006*), and the SC central element SYP-2, we also observed greater pairing and homologous synapsis in *pph-4.1; dsb-1(5A)* double mutants compared to *pph-4.1* single mutants (*Figure 3—figure supplement 1*). Previously, we have shown that the *pph-4.1* mutation leads to non-homologous synapsis including both fold-over synapsis within a single chromosome and promiscuous synapsis between non-homologous chromosomes (*Sato-Carlton et al., 2014*). To test whether the *dsb-1(5A)* allele improves homologous synapsis of *pph-4.1* mutants by changing the timing of synapsis initiation or completion, we performed immunofluorescence against the SC axial element protein HTP-3 and the central region protein SYP-2 in whole gonads of wild type, *pph-4.1*, and *pph-4.1; dsb-1(5A)*, and found no difference in the timing of synapsis (*Figure 3—figure supplement 1*). Similarly, heterozygous mutation of *atl-1* led to increased DSB levels and improved homologous synapsis in a *pph-4.1* background (*Figure 3—figure supplement 1*). Consistent with other recent discoveries that synapsis prior to recombination in *C. elegans* is a dynamic state that later becomes stabilized by recombination (*Liu et al., 2021*; *Machovina et al., 2016*; *Nadarajan et al., 2017*; *Pattabiraman et al., 2017*; *Roelens et al., 2015*), these results indicate that introduction of DSBs into a *pph-4.1* mutant also leads to increased fidelity of homologous pairing and synapsis.

We verified the formation of bivalents in *pph-4.1; dsb-1(5A)* animals by counting the number of DAPI-stained bodies in oocytes at diakinesis (*Figure 3C and E*). In the wild type, nearly 100% of diakinesis nuclei show six DAPI-stained bodies corresponding to six bivalents, while most nuclei show univalents in *pph-4.1* single mutants (*Sato-Carlton et al., 2014*). As expected from the viability rescue and the increase in homologous pairing and synapsis, the number of bivalents in *pph-4.1; dsb-1(5A)* mutants was significantly higher than in *pph-4.1* single mutants.

The apparent rescue of embryonic viability by increased DSBs in *pph-4.1; dsb-1(5A)* mutants contrasts with our previous observation that γ-ray irradiation (10 Gy) did not rescue bivalent formation in *pph-4.1* mutants (*Sato-Carlton et al., 2014*). However, at a higher dose of γ-irradiation (50 Gy), we observed rescue of bivalent formation in *pph-4.1* mutants, with 25% of oocytes showing six bivalents (*Figure 3D and E*) as part of an overall shift toward higher numbers of bivalents. This rescue in bivalent formation is not as high as that seen in *pph-4.1; dsb-1(5A)* mutants, although 50 Gy of γ-rays leads to more RAD-51 foci than the 5A allele (data not shown). Since 10 Gy irradiation at least partially rescues bivalent formation in mutants carrying the null *spo-11(me44)* allele, but does not rescue *pph-4.1* mutants (*Dernburg et al., 1998*; *Sato-Carlton et al., 2014*), our observation suggests that conversion from DSBs to crossovers is not as efficient in *pph-4.1* mutants, and thus *pph-4.1* mutants require more DSBs than *spo-11* mutants for bivalent formation. This requirement for higher break numbers is likely imposed by the combined deficiencies of *pph-4.1* mutants both in timely processing of recombination intermediates as well as in preventing non-homologous synapsis.

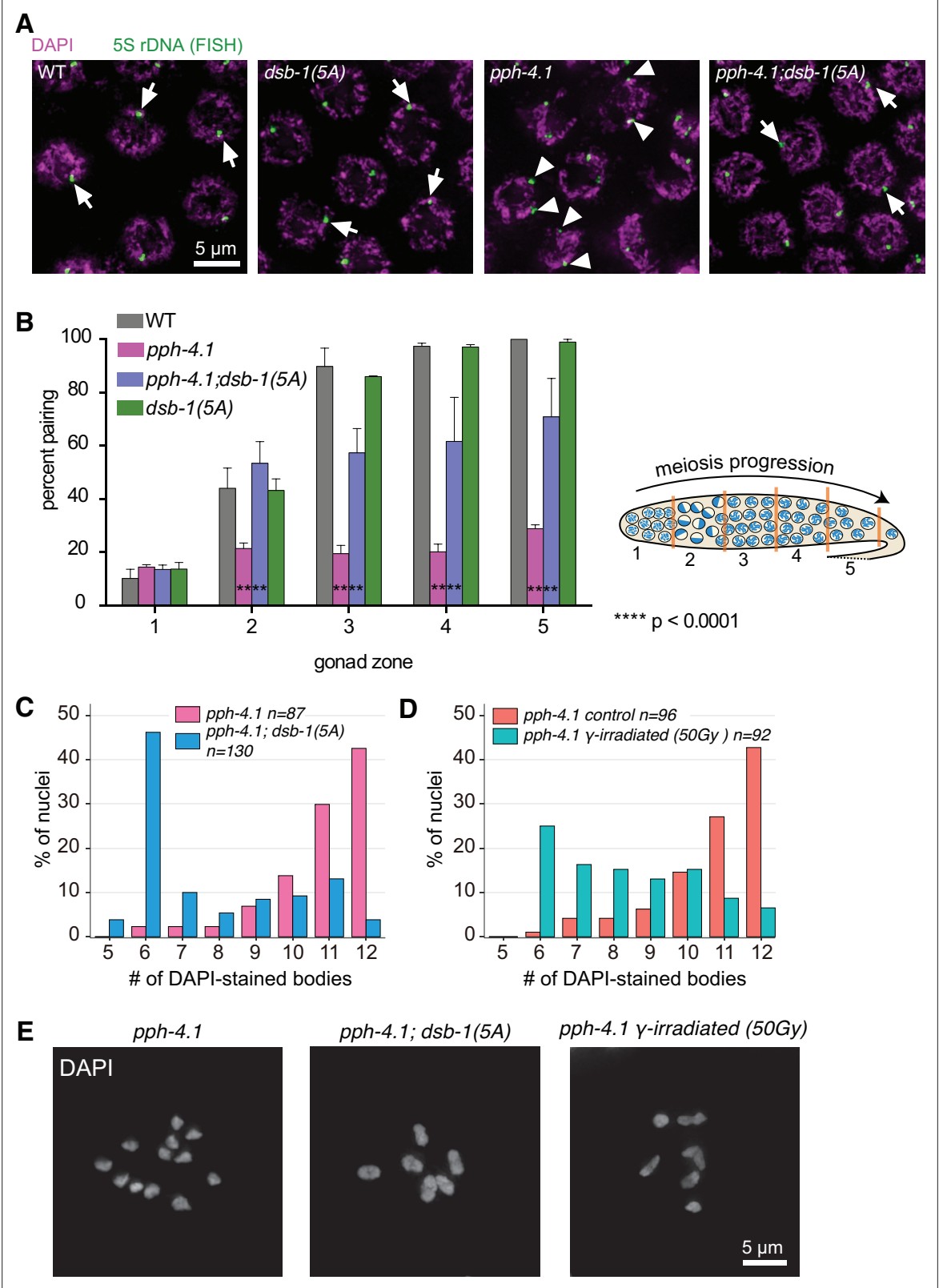

**Figure 3.** Chiasma formation and homologous pairing defects are partially rescued by the *dsb-1(5A)* allele in *pph-4.1* mutants. (**A**) FISH images show paired 5S rDNA sites in wild type, *dsb-1(5A)* and *pph-4.1(tm1598); dsb-1(5A)* worms (arrows indicate paired foci), and unpaired sites in *pph-4.1(tm1598)* mutants (arrowheads indicate unpaired foci) at pachytene. Scale bar, 5 µm. (**B**) *Left:* Quantification of pairing for chromosome V shown as the percent of nuclei with paired signals in each zone. Data are presented as mean ± SEM, two gonads were scored for wild type; three gonads were scored for

*Figure 3 continued on next page*

**Figure 3 continued**

*dsb-1(5A)*; four gonads were scored for *pph-4.1(tm1598)* and *pph-4.1(tm1598); dsb-1(5A)*. The total number of nuclei scored for zones 1–5 respectively was as follows: for wild type, 111, 150, 125, 95, 32; for *dsb-1(5A)*, 181, 228, 222, 143, 59; for *pph-4.1(tm1598)*, 300, 283, 257, 219, 124; for *pph-4.1(tm1598); dsb-1(5A)*, 335, 318, 266, 224, 137. Significance was assessed by chi-squared test for independence, ****p<0.0001. *Right:* Schematic showing a hermaphrodite gonad divided into five equally sized zones for FISH focus scoring (**Figure 3—source data 1**). (**C**) The number of DAPI-stained bodies shown as percentages of the indicated number of diakinesis oocyte nuclei scored for *pph-4.1(tm1598)* and *pph-4.1(tm1598); dsb-1(5A)* mutants. The numbers of nuclei scored for each genotype were: 87 for *pph-4.1(tm1598)*, 130 for *pph-4.1(tm1598); dsb-1(5A)* (**Figure 3—source data 2**). (**D**) The number of DAPI-stained bodies shown as percentages of the indicated number of diakinesis oocyte nuclei scored in *pph-4.1(tm1598)* mutants with or without 50 Gy γ-irradiation. The numbers of nuclei scored for each genotype were: 96 for *pph-4.1(tm1598)* control, 92 for *pph-4.1(tm1598)* γ-irradiated (**Figure 3—source data 3**). (**E**) Images of DAPI-stained diakinesis nuclei in a *pph-4.1(tm1598)* mutant and a *pph-4.1(tm1598); dsb-1(5A)* double mutant, as well as a *pph-4.1(tm1598)* mutant exposed to 50 Gy of γ-irradiation. Scale bar, 5 μm.

The online version of this article includes the following source data and figure supplement(s) for figure 3:

**Source data 1.** Number of paired and unpaired nuclei in the genotypes indicated in **Figure 3B**.

**Source data 2.** Number of nuclei with indicated DAPI body numbers of each genotype in **Figure 3C**.

**Source data 3.** Number of nuclei with indicated DAPI body numbers of each genotype in **Figure 3D**.

**Figure supplement 1.** Synapsis in *pph-4.1; dsb-1(5A)* and *pph-4.1; atl-1/nT1* mutants.

## DSB-1 non-phosphorylatable mutants rescue the defects of dsb-2 mutants

Many species in the *Caenorhabditis* genus possess a paralog of *dsb-1*, called *dsb-2*. In *C. elegans dsb-2(me96)* null mutants, a profound reduction of DSBs leads to crossover defects, causing severe embryonic inviability that increases with maternal age (**Rosu et al., 2013**). However, DSBs are not completely eliminated in *dsb-2* mutants, suggesting that while DSB-1 activity in the absence of DSB-2 can suffice to initiate DSBs, this activity is lower compared to when DSB-2 is present. The ability of the non-phosphorylatable *dsb-1(5A)* allele to rescue the loss of DSBs in *pph-4.1* mutants suggests that the 5A allele is hyperactive and not subject to downregulation through phosphorylation. To test whether this hyperactive allele depends on DSB-2 for its high levels of break formation, we performed RAD-51 immunofluorescence on *dsb-1(5A); dsb-2* germlines. In agreement with previous observations, we found very few RAD-51 foci in *dsb-2* mutant nuclei. However, the *dsb-1(5A)* allele strongly rescued DSB formation in the *dsb-2* null background to levels higher than wild type, similar to what is seen in the *dsb-1(5A)* allele alone (**Figure 4A and B**). Thus, non-phosphorylatable DSB-1 is capable of promoting high levels of DSBs in the complete absence of DSB-2 protein, providing further evidence of the 5A allele's hyperactivity. Moreover, this increased number of DSBs leads to normal bivalent formation and production of fully viable embryos in *dsb-2; dsb-1(5A)* double mutants (**Figure 4C**, **Table 1**), consistent with the rescue of viability of *dsb-2* mutants by γ-rays (**Rosu et al., 2013**). These results further show that the loss of DSBs in the *dsb-2* mutant depends on some or all of the five SQ sites in DSB-1, providing further evidence that phosphorylation of those sites leads to DSB loss.

The phosphorylation motifs of DSB-1 differentially rescue dsb-2 and pph-4.1.

To gain further insight into the five SQ sites, we generated a series of *dsb-1* non-phosphorylatable mutants: *dsb-1(1A)*, which is *dsb-1(S186A); dsb-1(2A)*, which is *dsb-1(S137A_S143A)* and *dsb-1(3A)*, which is *dsb-1(S137A_S143A_S186A)* based on the observation that S137 and S186 are highly conserved within the *elegans* group of *Caenorhabditis* (**Figure 5A**, **Figure 2—figure supplement 1**), and examined if they rescue *dsb-2* mutants. We found that the both *dsb-1(1A)* and *dsb-1(3A)* alleles were sufficient to rescue embryonic viability to wild-type levels at all maternal ages (**Figure 5B**, left; **Table 1**). In contrast, the *dsb-1(2A)* mutation, which does not include S186, rescued the embryonic viability defect in young *dsb-2* adults (day 1 post-L4 larval stage), but the rescue was less pronounced in older animals (day 3 post-L4) (**Figure 5B,** left). We therefore conclude that phosphorylation of any of the SQ motifs in the intrinsically disordered region of DSB-1 may contribute to shutting down the DSB-promoting activity of DSB-1, with S186 phosphorylation likely to be a strong determinant of reduced DSB activity in aged animals. Consistent with this, quantification of RAD-51 foci in *dsb-1 (1A), (2A), (3A)*, and *(5A)* mutants revealed that the mutants containing S186A (1A, 3A, and 5A mutants) showed an increase in DSB levels either in early prophase (zone 3 for 1A and 3A) or throughout prophase (for 5A) compared to the wild type (**Figure 5—figure supplement 1**). We also found that 5A mutants showed an even greater number of DSBs than 3A mutants, suggesting that the SQ motifs

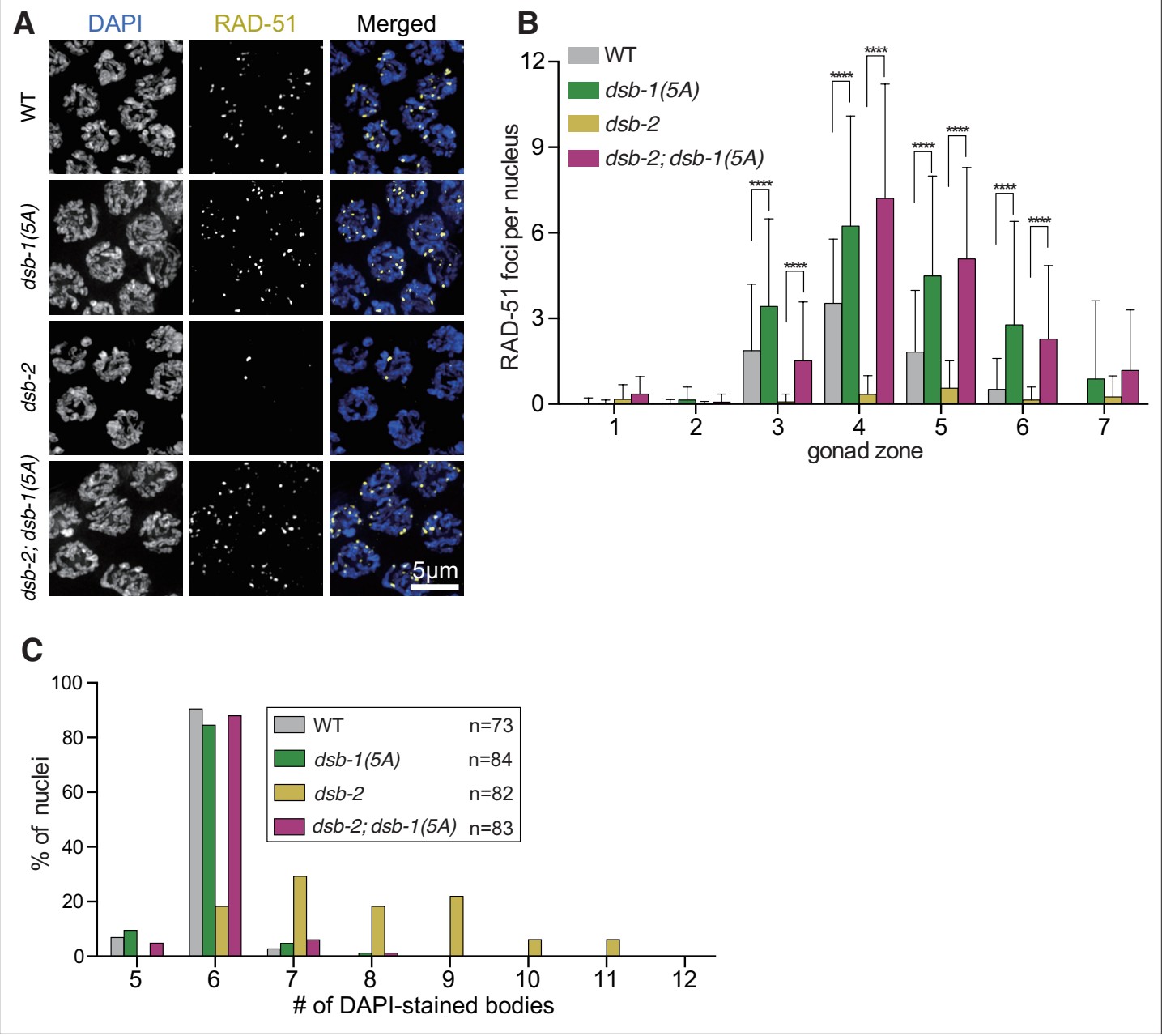

**Figure 4.** The *dsb-1(5A)* mutation rescues double-strand break (DSB) and crossover formation in *dsb-2* mutants. (**A**) Immunofluorescence images of RAD-51 foci in mid-pachytene nuclei of the indicated genotypes. Scale bar, 5 μm. (**B**) Quantification of RAD-51 foci in the gonads of the genotypes indicated in (**A**). Data are presented as mean ± SEM; three gonads were scored in *dsb-2(me96)*, *dsb1(5A)*, and *dsb-2(me96); dsb-1(5A)*, respectively, and two gonads were scored in wild type; the numbers of nuclei scored in zones 1–7 were as follows: for wild type, 57, 99, 112, 123, 109, 62, 27; for *dsb-2(me96)*, 124, 105, 108, 114, 108, 84, 53; for *dsb-1(5A)*, 126, 119, 103, 118, 116, 101, 79; for *dsb-2(me96); dsb-1(5A)*, 94, 131, 118, 91, 93, 96, 71. Significance was assessed via two-tailed *t* test, ****$p < 0.0001$ (*Figure 4—source data 1*). (**C**) The number of DAPI-stained bodies shown as percentages of the indicated number of diakinesis oocyte nuclei scored for each genotype. The numbers of nuclei scored for each genotype were: 73 for wild type, 84 for *dsb-1(5A)*, 82 for *dsb-2(me96)*, 83 for *dsb-2(me96); dsb-1(5A)* (*Figure 4—source data 2*).

The online version of this article includes the following source data for figure 4:

**Source data 1.** RAD-51 foci numbers graphed in *Figure 4B*.

**Source data 2.** Number of nuclei with indicated DAPI body numbers of each genotype in *Figure 4C*.

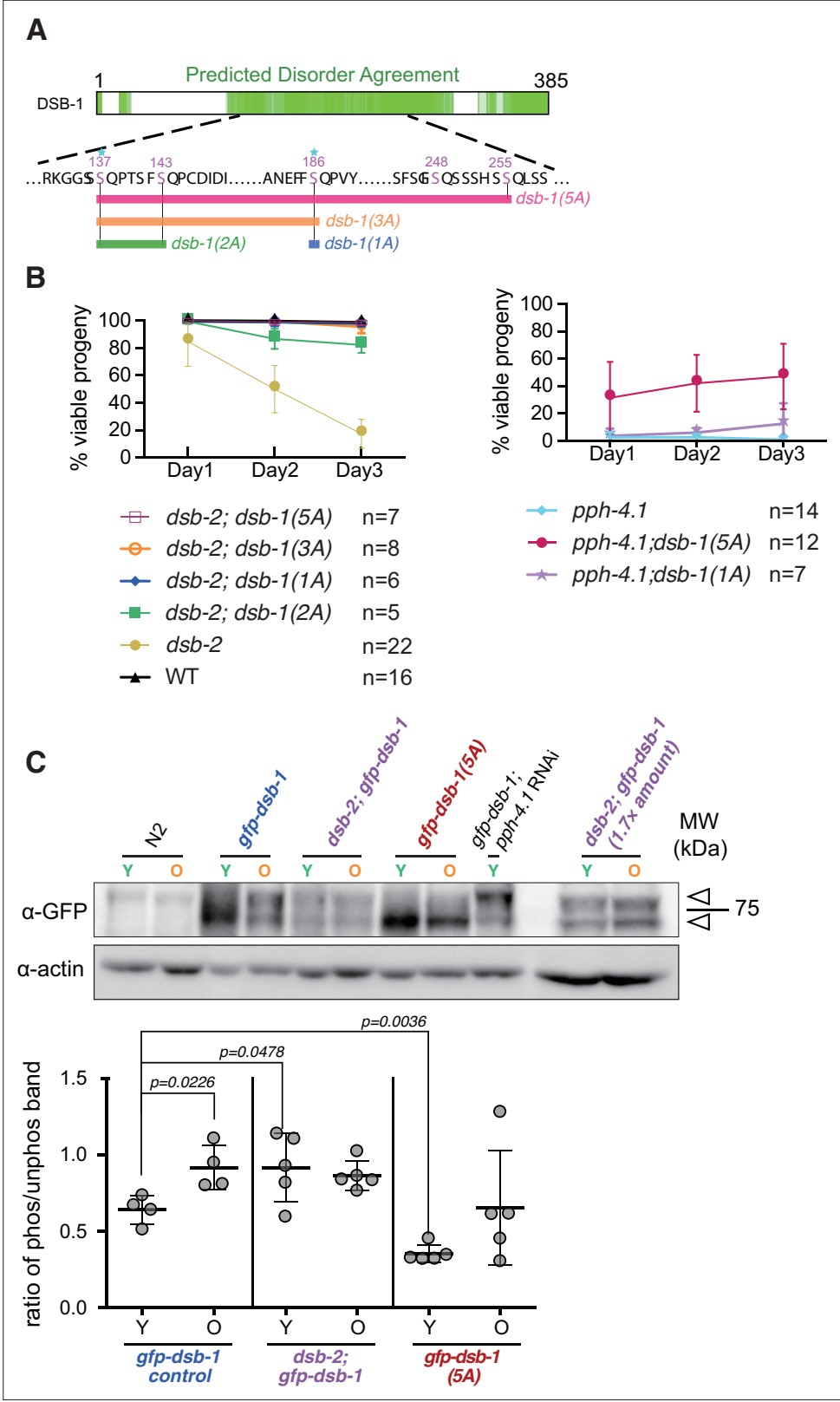

**Figure 5.** Alanine substitution of serine 186 in DSB-1 suffices to rescue the *dsb-2* mutation. (**A**) Diagram depicting a series of *dsb-1* phospho mutants: *dsb-1(1A)* is *dsb-1(S186A)*; *dsb-1(2A)* is *dsb-1(S137A; S143A)*; *dsb-1(3A)* is *dsb-1(S137A; S143A; S186A)* and *dsb-1(5A)* is *dsb-1(S137A; S143A; S186A; S248A; S255A)*. (**B**) The frequency of viable embryos from eggs laid by hermaphrodites of the indicated genotypes during the indicated time interval

*Figure 5 continued on next page*

*Figure 5 continued*

after the L4 larval stage. Data are presented as mean ± SEM from at least five biological replicates (*Figure 5—source data 1*). (**C**) *Top:* Western blots of GFP-fused DSB-1 from young adults (Y, 24 hr post-L4 larval stage) and old adults (O, 72 hr post-L4 larval stage) of the indicated genotypes, probed with α-GFP; arrowheads indicate the two GFP-DSB-1- specific bands in the blot. A total protein amount of 97 µg was loaded in each lane except for the two lanes of *dsb-2; gfp-dsb-1* double mutants on the right, in which 162 µg protein was loaded in each lane. Loading controls (α-actin) are shown at bottom. *Bottom:* Quantified band intensity ratio of phos-GFP-DSB-1 to non-phos-GFP-DSB-1 in the indicated genotypes. Data are presented as mean ± SD from at least two biological replicates. Significance was assessed via two-tailed t test with Welch's correction (*Figure 5—source data 2*, *Figure 5—source data 2*).

The online version of this article includes the following source data and figure supplement(s) for figure 5:

**Source data 1.** Number of daily laid viable progeny of the indicated genotypes in *Figure 5B*.

**Source data 2.** Intensity of phos-GFP-DSB-1 and non-phos-GFP band in *Figure 5C*.

**Source data 3.** Western blotting raw images in *Figure 5C*.

**Figure supplement 1.** Double-strand break (DSB) formation in a series of *dsb-1* non-phosphorylatable mutants.

**Figure supplement 1—source data 1.** RAD-51 foci numbers graphed in *Figure 5—figure supplement 1*.

**Figure supplement 2.** Double-strand break (DSB) formation and synapsis in *pph-4.1; dsb-1(1A)* and *dsb-2; dsb-1(1A)* mutants.

**Figure supplement 2—source data 1.** RAD-51 foci numbers graphed in *Figure 5—figure supplement 2B*.

at S248 and S255, which are mutated in 5A but not in 3A, contribute to shutting down the activity of DSB-1 in an additive manner (*Figure 5—figure supplement 1*).

While the *dsb-1(1A)* allele fully rescued *dsb-2* mutants, it did not rescue embryonic viability in *pph-4.1* mutants (*Figure 5B*, right; *Table 1*). Gonads of *dsb-1(1A)* mutants showed mildly increased levels of RAD-51 foci compared to wild type, and introducing the *dsb-1(1A)* mutation increased RAD-51 foci both in *pph-4.1; dsb-1(1A)* and in *dsb-2; dsb-1(1A)* mutants compared to the respective single *pph-4.1* or *dsb-2* mutants (*Figure 5—figure supplement 2A,B*). The difference in embryonic viability likely reflects inefficient processing of DSBs to COs in *pph-4* mutants, similar to the prior observation that high levels of γ-irradiation are required to rescue bivalent formation in this mutant.

We and others have shown that DSB initiation activity is significantly reduced in older adults (over 72 hr post-L4) compared to young adults (24 hr post-L4) in *rad-54(ok617)* mutants, in which DSBs are generated but the resulting recombination intermediates are trapped without completion of repair (*Raices et al., 2021*; *Sato-Carlton et al., 2014*). We therefore wondered whether DSB-1 may become more phosphorylated with age, thereby contributing to age-dependent reduction of DSB activity. To test this, we performed western blotting to assess DSB-1 phosphorylation levels in young (24 hr post-L4) versus older (72 hr post-L4) animals. Protein extracts were made from either wild type, *gfp-dsb-1* or *gfp-dsb-1(5A)* worms in varying combination with *dsb-2* and 24 hr treatment with *pph-4.1* RNAi, blotted onto membranes and probed with α-GFP antibodies. In *gfp-dsb-1* animals, the proportion of a slow-migrating band significantly increased in older animals (*Figure 5C*), demonstrating an increased proportion of phosphorylated DSB-1. An overall reduced amount of GFP-DSB-1 was detected in the *dsb-2* background, consistent with previous studies (*Hinman et al., 2021*; *Rosu et al., 2013*; *Stamper et al., 2013*). The phosphorylated band was proportionally stronger in young *dsb-2* mutants compared to young control animals, suggesting that both reduced protein levels and increased phosphorylation on DSB-1 likely contribute to lower DSB activity in *dsb-2* young animals (*Figure 5C*). However, somewhat surprisingly, the proportion of phosphorylated versus unphosphorylated GFP-DSB-1 did not change with age in *dsb-2* mutants. We verified this by loading a higher amount of protein and comparing the density of phosphorylated versus unphosphorylated bands in *dsb-2; gfp-dsb-1* (rightmost two lanes of *Figure 5C*). This raises the possibility that something other than DSB-1 phosphorylation may contribute to age-dependent loss of DSB production in *dsb-2* older animals. Alternatively, phosphorylation specifically on S186 may increase with age in *dsb-2* mutants, but this change may not be detectable without S186 phos-specific antibodies. We attempted to generate specific antibodies against DSB-1 phospho-S186 twice, but failed to obtain a specifically staining antibody. In *gfp-dsb-1(5A)* animals, the phosphorylated band proportion is strongly reduced both in young and old animals compared to the control. While the slow-migrating band is greatly reduced,

a smearing of GFP-DSB-1(5A) can be seen above the main band (*Figure 5C*). DSB-1 is a Ser/Thr-rich protein: possessing 91 residues of mostly Ser/Thr and some Tyr in the length of total 385 amino acids, and non-SQ serines and/or threonines make up almost 20% of the protein. The smearing of GFP-DSB-1(5A) suggests that phosphorylation may occur at some of the other 86 serines, threonines, or tyrosines, in the absence of phosphorylation at the five SQ sites. Taken together, our results suggest that age-dependent increase of DSB-1 phosphorylation contributes to a reduction in DSB levels.

## DSB-1 is predicted to form a heterotrimeric complex with DSB-2 and DSB-3

The recent identification of *dsb-3* as a nematode ortholog of Mei4 and its likely participation in a complex with DSB-1 and DSB-2 (*Hinman et al., 2021*) akin to the heterotrimeric Rec114-Mei4 complex of yeast (*Claeys Bouuaert et al., 2021*) prompted us to examine predicted structural properties of such a complex using the AlphaFold structure prediction pipeline (*Jumper et al., 2021*; *Mirdita et al., 2021*). Since both DSB-1 and DSB-2 are orthologs of Rec114, we tested three possible complexes: a fully heterotrimeric complex of DSB-1, DSB-2, and DSB-3, and complexes containing two copies of either DSB-1 or DSB-2 with one copy of DSB-3. We also generated predictions for the DSB-1, -2, and -3 orthologs from the closely related species *Caenorhabditis inopinata*, as well as for human and yeast Rec114-Mei4 complexes in 2:1 stoichiometry. In all predicted DSB-1:DSB-2:DSB-3 trimers, the alpha-helical C-termini of DSB-1 and DSB-2 wrap around each other to form a channel which accommodates the helical N-terminus of DSB-3; a similar structure was predicted for trimers of two Rec114 and one Mei4 (*Figure 6A*). In all species (nematode, yeast, and human), these independent structural predictions are in agreement with previous models based on yeast two-hybrid (*Hinman et al., 2021*; *Maleki et al., 2007*) and crosslinking mass spectrometry analysis (*Claeys Bouuaert et al., 2021*). This trimer prediction was not found in models of a DSB-2:DSB-2:DSB-3 trimer, but was found in three out of five DSB-1:DSB-1:DSB-3 models. These predictions raise the possibility that DSB-1 is more likely to bind DSB-3 in the absence of DSB-2, than DSB-2 is to bind DSB-3 in the absence of DSB-1. This asymmetry would be consistent with the more severe phenotype of *dsb-1* compared to *dsb-2* mutants, as well as with yeast two-hybrid evidence showing DSB-1, but not DSB-2, directly binds to DSB-3 (*Hinman et al., 2021*). The consistency of the structural prediction in three highly diverged species, and its agreement with known in vivo data, is highly suggestive of a conserved interaction. However, since the interacting regions within the predicted trimer does not involve the disordered domain of DSB-1 or any of its SQ motifs, this prediction does not suggest whether or how phosphorylation of DSB-1 might interfere with its ability to bind to the other members of this complex.

In summary, we have shown that DSB-1 activity is regulated by its phosphorylation levels, which are set by the opposing activities of PP4 phosphatase and ATR kinase (*Figure 6B*). We propose that this phosphoregulation ensures that not too many but not too few DSBs are generated on every chromosome to enable crossover formation and correct segregation of chromosomes in meiosis.

## Discussion

In this work, we have identified DSB-1 as a nexus of DSB initiation control by phosphoregulation in *C. elegans*. While regulation of the Spo11 cofactor Rec114 by the DNA damage kinases ATM[Tel1] and ATR[Mec1] has been previously investigated (*Carballo et al., 2013*), we show here for the first time that PP4 phosphatase plays an opposing role, working through DSB-1 to promote a number of breaks sufficient to generate a crossover on every chromosome. Moreover, we show that ATR kinase plays a key inhibitory role in *C. elegans* DSB initiation. Mutation of the DSB-1 ATM/ATR consensus site serines to alanine leads to increased DSB numbers in a wild-type background, and rescues the DSB loss phenotypes of *pph-4.1* and *dsb-2* mutants. This finding may illuminate a more straightforward mechanism of action compared to the yeast homolog Rec114, in which mutation of ATM/ATR consensus sites show contrasting results on DSB number depending on the assay used (*Kar and Hochwagen, 2021*; discussed in *Lukaszewicz et al., 2018*). Based on our results, the most straightforward model is that PPH-4.1 dephosphorylates DSB-1 to promote DSBs, whereas break-activated ATL-1 phosphorylates DSB-1 to prevent excess DSB production, and this phosphoregulation circuit ensures termination of DSB production only after sufficient numbers of DSBs are generated. This is consistent with the previous observations that many ATM/ATR targets such as Hop1, Mek1, RPA2, H2A, and Zip1 are

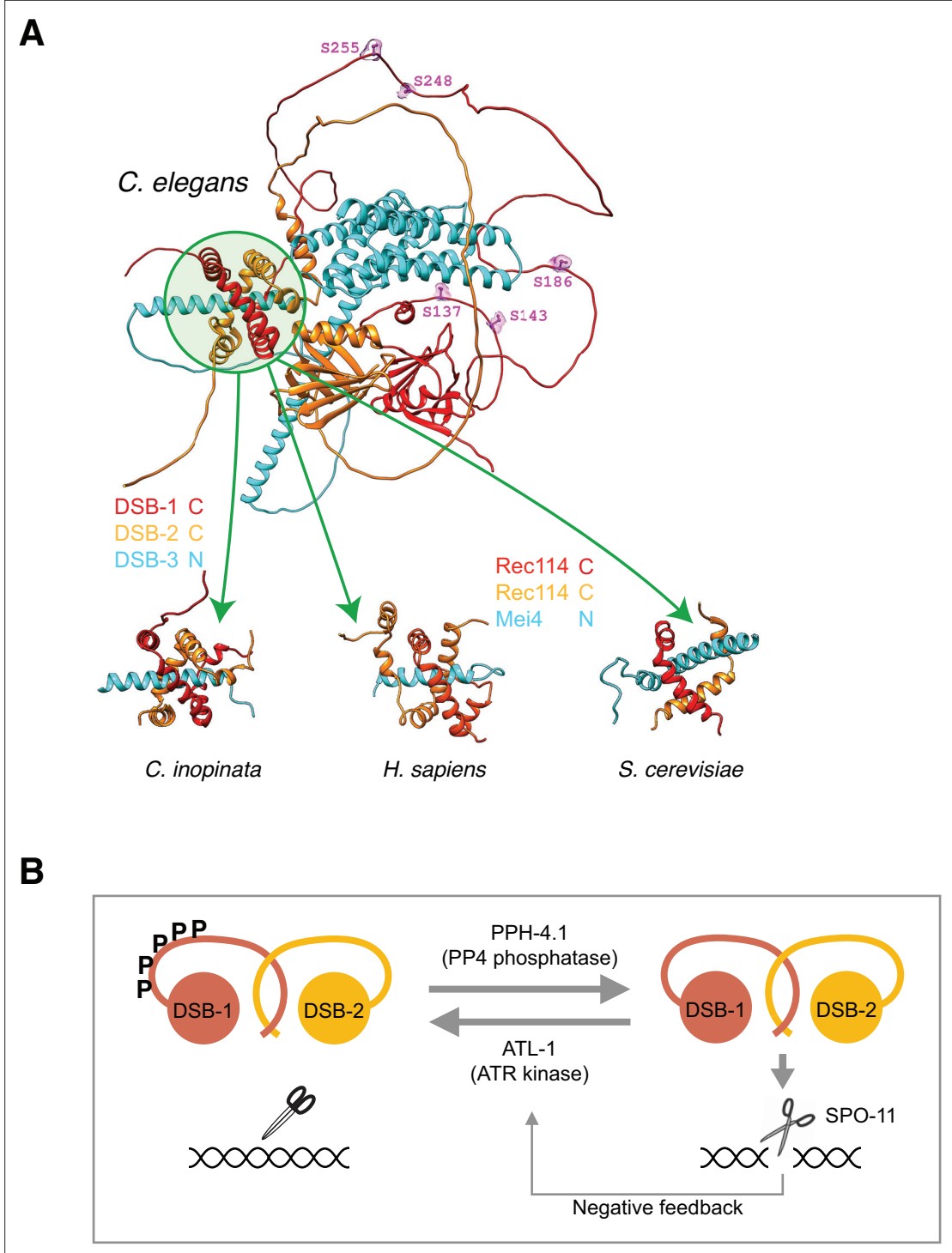

**Figure 6.** Structural prediction of double-strand break (DSB) factors and their interaction. (**A**) A representative structure of the DSB-1, DSB-2, and DSB-3 heterotrimer predicted by the AlphaFold structure prediction pipeline (*Jumper et al., 2021*; *Mirdita et al., 2021*) is shown at top. Green circle highlights the region predicted to be the trimerization interface containing the C-termini of DSB-1 and -2, and the N-terminus of DSB-3. ATM/ATR kinase phosphorylation consensus sites in the predicted disordered loop of DSB-1 are shown in magenta and labeled. Sub-regions of similar structures predicted for orthologs of DSB-1, DSB-2, and DSB-3 in the *Caenorhabditis elegans* sister species *Caenorhabditis inopinata*, as well as the putative Rec114/Rec114/Mei4 heterotrimer in human and budding yeast, are shown below. In all cases a predicted N-terminal alpha-helix of the Mei4 ortholog (DSB-3) transfixes a channel formed by the predicted C-terminal helices of the Rec114 orthologs wrapping around each other (*Figure 6—source data 1*). (**B**) A model showing antagonistic action of ATL-1 and PPH-4.1 in DSB-1 regulation.

*Figure 6 continued on next page*

*Figure 6 continued*

The online version of this article includes the following source data and figure supplement(s) for figure 6:

**Source data 1.** AlphaFold prediction files and settings (ZIP).

**Figure supplement 1.** Phylogenetic prediction of double-strand break (DSB) factors.

dephosphorylated in a PP4-dependent manner in other organisms (*Chuang et al., 2012*; *Falk et al., 2010*; *Keogh et al., 2006*; *Lee et al., 2010*). However, since other factors including an effector kinase Rad53 (Chk2) are also known to be dephosphorylated in a PP4-dependent manner in budding yeast (*Villoria et al., 2019*), we cannot exclude the possibility that PPH-4.1 indirectly reduces DSB-1 phosphorylation by dephosphorylating upstream factors regulating DSB-1. In budding yeast and *Arabidopsis*, previous studies have shown that DSB levels were not reduced in their corresponding PP4 mutants (*Falk et al., 2010*; *Nageswaran et al., 2021*). Currently, it is unknown if PPH-4.1 function in DSB-1/Rec114 control is conserved in other model organisms.

Since the mechanism by which the RMM group of cofactors (including DSB-1/2) promotes Spo11 activity remains unknown, how phosphorylation of these factors negates their activity is also not clear. Recent work has shown that neutralizing a conserved basic patch of either Rec114 or Mer2 leads to loss of DNA binding (*Claeys Bouuaert et al., 2021*); analogously, DSB-1 interaction with DNA may be prevented by the negative charge of phosphorylation. The same work showed that RMM proteins form DNA-dependent condensates, which raises the possibility that phosphorylation of the intrinsically disordered region could electrostatically alter any phase separation propensity of DSB-1. Further investigation of DSB-1 and DSB-2 in vivo is necessary to determine the extent to which they resemble their yeast orthologs with regard to condensate formation.

The diverged functions of DSB-1 and DSB-2 raise the question of which one of the paralogs is closer to the ancestral gene. We constructed a phylogenetic tree based on the longest isoform of each set of genes and found that the duplication of a single Rec114 ortholog into the paralogs DSB-1 and DSB-2 occurred early within the genus *Caenorhabditis*. Based on our inferred gene tree, we estimate that the *C. elegans* DSB-1 and DSB-2 protein sequences have undergone 1.36 and 2.02 amino acid substitutions per site, respectively, since the gene duplication event (*Figure 6—figure supplement 1*). The mean number for all DSB-1 and DSB-2 orthologs since the gene duplication event is 1.03 and 1.66 amino acid substitutions per site, respectively. The lower number for DSB-1 suggests that DSB-1 retains the essential ancestral function common to all Rec114 orthologs, while DSB-2 has evolved to perform a slightly modified role. We have demonstrated an increase in the proportion of phosphorylated compared to unphosphorylated DSB-1 in *dsb-2* young animals (*Figure 5C*), and showed that the hyperactive *dsb-1(S186A)* mutant rescues loss of *dsb-2*. These results suggest that DSB-2 is required to counteract the phosphorylation and thus downregulation of DSB-1. Taken together with previous results showing that loss of *dsb-2* leads to reduction in DSB-1 protein levels (*Rosu et al., 2013*; *Stamper et al., 2013*), it is likely that DSB-2 has evolved to perform an auxiliary role in DSB formation, stabilizing DSB-1 proteins and compensating for DSB-1's gradual deactivation, and thereby extending the window of fertility. We hypothesize that either increasing phosphorylation of DSB-1 S186 with age, or age-correlated increase of another factor that impedes DSB formation by phosphorylated DSB-1, underlies this drop in DSB formation with age seen in *C. elegans*.

Our observation that the *dsb-1(5A)* allele as well as γ-irradiation can increase homologous pairing, synapsis, and bivalent formation in *pph-4.1* mutants (*Figure 3*; *Figure 3—figure supplement 1*) adds to growing evidence that while initial pairing and synapsis in *C. elegans* does not depend on DSBs, stabilization and (when necessary) correction of synapsis does. Interestingly, the 5A allele rescues *pph-4.1* to a greater degree than 50 Gy of γ-rays, even though the number of induced breaks in γ-irradiated *pph-4.1* worms is higher than that seen in *pph-4.1; dsb-1(5A)* (data not shown). This difference has several possible explanations: perhaps the mechanism that corrects promiscuous synapsis requires canonical SPO-11-catalyzed breaks rather than the more complex damage caused by γ-rays. Alternatively, hyperactivation of the DNA damage response by high levels of both single-strand break and DSB in γ-irradiated worms may not allow homologous synapsis to reach the level seen in *pph-4.1; dsb-1(5A)* animals, or may interfere with efficient conversion of recombination intermediates to COs. In a similar vein, γ-ray-induced desynapsis (*Couteau and Zetka, 2011*) may have unknown deleterious effects on pairing, synapsis, or crossover formation specifically in *pph-4.1* mutants, contributing

to a lower frequency of rescue. The promiscuous pairing and synapsis in *pph-4.1* mutants cannot be attributed solely to a low number of DSBs, since null mutants of both *spo-11* and *dsb-1* that completely lack DSBs, as well as *dsb-2* mutants with severely reduced DSBs, synapse homologously (***Dernburg et al., 1998***; ***Rosu et al., 2013***; ***Stamper et al., 2013***). We hypothesize that in the absence of PPH-4.1 activity, hyperphosphorylation of one or more additional substrates causes promiscuous synapsis in a low-DSB (i.e., immature SC) environment, but providing additional DSBs yields a higher degree of synaptic fidelity, perhaps via homologous recombination. Further, the number of DSBs required to rescue embryonic viability in *pph-4.1* mutants must be higher than that needed to rescue viability in *dsb-2*, since the *dsb-1(1A)* allele suffices to fully rescue mutations in *dsb-2*, but has little effect on *pph-4.1* (***Figure 5B***, ***Table 1***). This inconsistency may be resolved by noting that rescue of viability in *pph-4.1* mutants requires rescue of both promiscuous pairing and synapsis and of crossover failure, and non-homologous synapsis is still observed in *pph-4.1; dsb-1(1A)* mutants (***Figure 5—figure supplement 2C***). Since *pph-4.1; dsb-1(1A)* germlines show numbers of RAD-51 foci intermediate between *pph-4.1* and *pph-4.1; dsb-1(5A)* (***Figure 2C***, ***Figure 5—figure supplement 2B***), we hypothesize that the number of DSBs required to correct promiscuous pairing in *pph-4.1* mutants is higher than that needed to guarantee a crossover on each chromosome. Incomplete rescue of homologous pairing and incompletely penetrant phenotypes in other known PP4-dependent processes such as centrosome maturation and sperm production (***Han et al., 2009***; ***Sumiyoshi et al., 2002***) that are not rescued by DSBs are likely responsible for the remaining brood size and viability defects in *pph-4.1; dsb-1(5A)* double mutants (***Figure 2D***).

The fact that worms carrying the *dsb-1(5A)* mutation enjoy full viability, with brood size and male incidence nearly identical to wild type despite the roughly twofold higher number of RAD-51 foci observed, raises the question of what role negative control of DSBs is playing in *C. elegans*. To examine whether *dsb-1(5A)* could sensitize worms to external DNA damage, we have γ-irradiated control and *dsb-1(5A)* animals and assayed for embryonic inviability as an indicator of unrepaired DNA breaks. However, *dsb-1(5A)* mutants showed no difference from control animals in embryonic viability or brood size after 30 or 75 Gy irradiation (data not shown), suggesting that meiocytes have a large capacity to repair DSBs in excess over wild-type levels. A recent study examining the effects of loss of germline ATM in mice (***Lukaszewicz et al., 2021***) discovered an increased incidence of large deletions and other rearrangements at hotspots. Limiting the number of DSBs through DSB-1 phosphorylation could forestall such mutagenic events and maintain genome integrity over generations. Further experiments analyzing long-term genome integrity in non-phosphorylatable *dsb-1* mutants are needed to address this issue. While the physiological role of DSB-1 phosphorylation remains unclear, our work provides the first evidence that the ability of PPH-4.1 to regulate DSB-1 phosphorylation levels in meiotic prophase is critical to provide a sufficient number of breaks to ensure chiasma formation on each chromosome pair.

## Materials and methods

**Key resources table**

| Reagent type (species) or resource | Designation | Source or reference | Identifiers | Additional information |
|---|---|---|---|---|
| Gene (*Caenorhabditis elegans*) | dsb-1 | WormBase/***Stamper et al., 2013*** PMID:23990794 | WormBase ID:WBGene00008580 | |
| Gene (*Caenorhabditis elegans*) | pph-4.1 | WormBase/***Sato-Carlton et al., 2014*** PMID:25340746 | WormBase ID:WBGene00004085 | |
| Gene (*Caenorhabditis elegans*) | atl-1 | WormBase | WormBase ID:WBGene00000226 | |
| Gene (*Caenorhabditis elegans*) | atm-1 | WormBase | WormBase ID:WBGene00000227 | |

*Continued on next page*

*Continued*

| Reagent type (species) or resource | Designation | Source or reference | Identifiers | Additional information |
|---|---|---|---|---|
| Gene (*Caenorhabditis elegans*) | *dsb-2* | WormBase/*Rosu et al., 2013* PMID:23950729 | WormBase ID:WBGene00194892 | |
| Strain, strain background (*Escherichia coli*) | *pph-4.1* RNAi-L4440 in HT115 | *Sato-Carlton et al., 2014* PMID:25340746 | | |
| Strain, strain background (*Escherichia coli*) | L4440 in HT115 | Ahringer Lab RNAi library (Source Biosciences) | | |
| Strain, strain background (*Caenorhabditis elegans*) | For *C. elegans* allele and strain information, see *Supplementary file 1* | This paper | N/A | See *Supplementary file 1* |
| Genetic reagent (*Caenorhabditis elegans*) | For CRISPR/Cas9 reagents, see sequence-based reagent and peptide, recombinant protein. | This paper | N/A | Purchased from IDT |
| Antibody | Anti-GFP (mouse monoclonal) | Santa Cruz | Cat#sc-9996 | WB (1:1000) |
| Antibody | Anti-Actin (rabbit polyclonal) | Santa Cruz | Cat#sc-1615 | WB (1:3000) |
| Antibody | Anti-RAD-51(rabbit polyclonal) | SDIX/Novus Biologicals | Cat#29480002 lot#G3048-009A02 | IF (1:10,000) |
| Antibody | Anti-DSB-1 (guinea pig polyclonal) | *Stamper et al., 2013* PMID:23990794 | N/A | WB (1:75) |
| Antibody | Anti-ZIM-3 (rabbit polyclonal) | *Phillips and Dernburg, 2006* PMID:17141157 | N/A | IF (1:2000) |
| Antibody | Anti-SYP-1 (guinea pig polyclonal) | *Sato-Carlton et al., 2020* PMID:33175901 | N/A | IF (1:100) |
| Antibody | Anti-SYP-2 (rat polyclonal) | *Sato-Carlton et al., 2020* PMID:33175901 | N/A | IF (1:200) |
| Antibody | Anti-HTP-3 (guinea pig polyclonal) | *MacQueen et al., 2005* PMID:16360034 | N/A | IF (1:500) |
| Antibody | Alexa488-anti-rabbit (donkey polyclonal) | Jackson ImmunoResearch | Cat#711-545-152, lot#109880 | IF (1:500) |
| Antibody | DyLight649-anti-guinea pig (donkey polyclonal) | Jackson ImmunoResearch | Cat#706-495-148, lot#95544 | IF (1:500) |
| Antibody | DyLight594-anti-guinea pig (donkey polyclonal) | Jackson ImmunoResearch | Cat#706-515-148, lot#94259 | IF (1:500) |
| Antibody | DyLight649-anti-rat (donkey polyclonal) | Jackson ImmunoResearch | Cat#712-495-153, lot#94218 | IF (1:500) |
| Antibody | HRP-conjugated anti-mouse (sheep polyclonal) | GE Healthcare BioSciences | Cat#NIF825 | WB (1:10,000) |
| Antibody | HRP-conjugated anti-rabbit (goat polyclonal) | Abcam | Cat#ab97051 | WB (1:10,000) |
| Antibody | HRP-conjugated anti-guinea pig (goat polyclonal) | Beckman Coulter | Cat#732868 | WB (1:10,000) |
| Sequence-based reagent | FISH probe to the right arm of Chromosome V (5S rDNA) | *Dernburg et al., 1998* PMID:9708740 | N/A | |
| Sequence-based reagent | Alt-R CRISPR-Cas9 tracrRNA | IDT | Cat# 1072532 | |
| Sequence-based reagent | *dpy-10* crRNA: 5'-GCTACCATAGGCACCACGAG-3' | https://www.ncbi.nlm.nih.gov/pubmed/25161212 | N/A | |

*Continued on next page*

*Continued*

| Reagent type (species) or resource | Designation | Source or reference | Identifiers | Additional information |
|---|---|---|---|---|
| Sequence-based reagent | dpy-10(cn64) homology template for CRISPR 5'-cacttgaacttcaatacggcaagatgagaatgactggaaaccgta ccgcATgCggtgcctatggtagcggagcttcacatggcttcagaccaacagcct-3' | https://www.ncbi.nlm.nih.gov/pubmed/25161212 | N/A | |
| Sequence-based reagent | For crRNAs, repair templates and genotyping primers, see ***Supplementary file 2*** | This paper | N/A | Purchased from IDT |
| Peptide, recombinant protein | Recombinant Cas9 protein | UC Berkeley QB3 Macrolab | https://macrolab.qb3.berkeley.edu/cas9-nls-purified-protein/ | |
| Commercial assay or kit | Pierce BCA Protein Assay Kit | ThermoFisher Scientific | Cat#23227 | |
| Chemical compound, drug | DAPI (4',6-Diamidino-2-phenylindole dihydrochloride) | Nacalai Inc | Cat#11034–56 | |
| Chemical compound, drug | Nuclease-free Duplex Buffer | IDT | Cat#11-01-03-01 | |
| Chemical compound, drug | GFP-Trap magnetic beads | ChromoTek | Cat#gtma-20 | |
| Chemical compound, drug | Lambda Protein Phosphatase | BioLabs | Cat#P0753S | |
| Chemical compound, drug | Chemilumi-one super | Nacalai Inc | Cat#02230–30 | |
| Chemical compound, drug | Chemilumi-one ultra | Nacalai Inc | Cat#11644–24 | |
| Chemical compound, drug | skim milk | Nacalai Inc | Cat#31149–75 | |
| Chemical compound, drug | SuperSep (TM) Ace, 5%–12%, 13well | Wako | Cat#199–15191 | |
| Chemical compound, drug | NuPAGE 4% to 12%, Bis-Tris, 1.0 mm, Mini Protein Gel, 12-well | Invitrogen | Cat#NP0322BOX | |
| Software, algorithm | softWoRx suite | Applied Precision/GE Healthcare | N/A | |
| Software, algorithm | OMERO | ***Burel et al., 2015*** PMID:26223880 | https://www.openmicroscopy.org/omero/ | |
| Software, algorithm | Priism | ***Chen et al., 1996*** PMID:8742723 | https://msg.ucsf.edu | |
| Software, algorithm | Prism6 | GraphPad | https://www.graphpad.dom/scientific-software/prism | |
| Software, algorithm | Mafft v7.487 | ***Katoh and Standley, 2013*** PMID:23329690 | https://mafft.cbrc.jp/alignment/software/ | |
| Software, algorithm | Fiji | ***Schindelin et al., 2012*** PMID:22743772 | https://fiji.sc | |

*Continued on next page*

*Continued*

| Reagent type (species) or resource | Designation | Source or reference | Identifiers | Additional information |
|---|---|---|---|---|
| Software, algorithm | AGAT v0.4.0 | *Dainat et al., 2020* doi:10.5281/ZENODO.3552717 | https://github.com/NBISweden/AGAT | |
| Software, algorithm | OrthoFinder v2.5.4 | *Emms and Kelly, 2019, Emms, 2022* PMID:31727128 | https://github.com/davidemms/OrthoFinder | |
| Software, algorithm | FSA v1.15.9 | *Bradley et al., 2009* PMID:19478997 | http://fsa.sourceforge.net/ | |
| Software, algorithm | IQ-TREE v2.2.0-beta | *Nguyen et al., 2015* PMID:25371430 | http://www.iqtree.org/ | |
| Software, algorithm | ETE3 Python module | *Huerta-Cepas et al., 2016* PMID:26921390 | http://etetoolkit.org/ | |

## Worm strains and antibodies

*C. elegans* strains were maintained at 20°C on nematode growth medium (NGM) plates seeded with OP50 bacteria under standard conditions (*Brenner, 1974*). Bristol N2 was used as the wild-type strain and all mutants were derived from an N2 background. A list of all strains and antibodies used is provided in the Key resources table. All cytological experiments were performed on adult hermaphrodite germlines.

## Generation of mutants via CRISPR-Cas9 genome editing system

CRISPR-Cas9 genome editing using *dpy-10* as co-CRISPR marker (*Arribere et al., 2014*) was applied to generate *dsb-1* N-terminal tagged (*gfp-dsb-1*) and *dsb-1* non-phosphorylatable lines. A 10 μL mixture containing 17.5 μM trans-activating CRISPR RNA (tracrRNA)/crRNA oligonucleotides (targeting *dsb-1* and *dpy-10*) purchased from Integrated DNA Technologies (IDT, Coralville, IA), 17.5 μM Cas9 protein produced by the MacroLab at UC Berkeley, and 6 μM single-stranded DNA oligonucleotide purchased from IDT or 150 ng/μL double-stranded DNA generated from PCR as a repair template was injected into the gonads of 24 hr post-L4 larval stage N2 hermaphrodites. To prevent re-editing by the CRISPR-Cas9 machinery, silent mutations were introduced into the target gene *dsb-1*. For *gfp-dsb-1*, an additional linker sequence of 3× glycine was introduced between the target site and GFP-tag sequence. Dpy or Rol F1 animals (*dpy-10* mutation homozygous or heterozygous, respectively) were picked to individual plates to self-propagate overnight and then screened for successful edits by PCR and DNA sequencing. A list of oligonucleotides used is provided in *Supplementary file 2*.

## RNA interference

RNA interference (RNAi) was carried out by feeding N2 or *gfp-dsb-1* worms with the HT115 bacteria expressing either the empty RNAi vector L4440 obtained from the Ahringer Lab RNAi library (*Kamath et al., 2003*) or a *pph-4.1* RNAi plasmid (*Sato-Carlton et al., 2014*). Worms were first synchronized through starvation and grown to the L4 larval stage on new NGM plates with OP50 bacteria. L4 worms were collected in M9 (41 mM Na$_2$HPO$_4$, 15 mM KH$_2$PO$_4$, 8.6 mM NaCl, 19 mM NH$_4$Cl)+0.01% Tween buffer, washed three times with M9 buffer and distributed to RNAi plates. About 30 hr later, the worms became gravid and were harvested in M9 + 0.01% Tween buffer, washed three times with M9 buffer and bleached for no more than 3 min to obtain F1 embryos. Collected embryos were placed to fresh RNAi plates and grown until 24 hr after the L4 larval stage. For western blot analysis, worms were harvested and washed three times or more in M9 buffer and frozen in liquid nitrogen.

## Auxin-induced protein depletion in worms

Depletion of AID-tagged proteins in the *C. elegans* germline was performed as previously described (*Zhang et al., 2015*). Briefly, 1 mM auxin (IAA, Alfa Aesar #10556, Haverhill, MA) was added into the NGM agar just before pouring plates. *Escherichia coli* OP50 bacteria cultured overnight were

concentrated, supplemented with 1 mM auxin, and spread on plates. These auxin plates were stored at 4°C in the dark and used within a month. NGM plates supplemented with ethanol (0.25% v/v) were used as a control. To obtain synchronized worms, L4 hermaphrodites were picked from the maintenance plates. Auxin treatment was performed by transferring worms to auxin plates and incubating for the indicated time at 20°C. Young adult animals (24 hr post-L4) were dissected for immunofluorescence analyses.

## Lysate preparations and phosphatase assay

To prepare samples for general western blotting of GFP-DSB-1, frozen worm pellet was suspended in urea lysis buffer (20 mM HEPES pH 8.0, 9 M urea, 1 mM sodium orthovanadate), sonicated (Taitec VP505 homogenizer, Koshigaya City, Japan; 50% output power, cycle of 10 s on and 10 s off for 7 min or more until worm bodies were completely broken down by visual inspection) and spun down at 16,000 *g* at 4°C for 15 min. The supernatant was used to measure protein concentration using the BCA kit (Pierce BCA protein assay kit #23225; Thermo Scientific, Waltham, MA), and a set amount of protein was loaded for western blotting after boiling for 10 min in SDS-PAGE sample buffer.

For the phosphatase assay, frozen *gfp-dsb-1* worms treated with *pph-4* RNAi were suspended in lysis buffer (50 mM HEPES pH 7.0, 100 mM NaCl, 2 mM DTT, 0.1 mM EGTA) containing protease inhibitor cocktail (Nacalai #03969-21, Kyoto, Japan) and sonicated (Taitec VP505 homogenizer, Koshigaya City, Japan; 50% output power, cycle of 10 s on and 10 s off for 7 min). This lysate was first spun down for 100 *g* at 4°C for 3 min to remove worm debris. The supernatant was further spun down at 16,000 *g* at 4°C for 15 min to pellet nuclei, and the pellet was resuspended in RIPA buffer (150 mM NaCl, 1% Triton X-100, 0.5% sodium deoxycholate, 0.1% SDS (sodium dodecyl sulfate), 50 mM Tris pH 8.0) containing protease inhibitor cocktail (Nacalai #03969-21, Kyoto, Japan) and incubated at 4°C for 30 min to solubilize nuclear proteins. The lysate was further sonicated (Taitec VP505 homogenizer, Koshigaya City, Japan; 50% output power, cycle of 10 s on and 10 s off for 5 min) and used for the phosphatase assay (NEB lambda phosphatase #P0753S, Ipswich, MA) following the manufacturer's instructions at 30°C for 2 hr. Then the SDS-PAGE sample buffer was added to the phosphatase reaction, boiled at 95°C for 10 min and used for western blotting.

For endogenous DSB-1 immunoblotting experiments in *Figure 1C, D and E* and *Figure 1—figure supplement 1C* and D, 50 worms of 24 hr post-L4 stage were picked into M9 + 0.05% Tween for each lane and washed twice, then SDS-PAGE sample buffer was added to the harvested worms, and after boiling for 5 min at 95°C, the protein was flash centrifuged and loaded on the gel.

## Western blot

For western blotting of GFP-fused DSB-1, SDS-PAGE was carried out using 5–12% Wako gradient gel (Wako #199-15191, Tokyo, Japan), and proteins were transferred to a PVDF membrane at 4°C, 80 V for 2.5 hr. The membrane was blocked with TBST buffer (TBS and 0.1% Tween) containing 5% skim milk (Nacalai Inc #31149-75, Kyoto, Japan) at room temperature for 1 hr and probed with primary antibody solution containing 2.5% skim milk at 4°C overnight followed by additional 2 hr at room temperature, washed with TBST for four times, probed with secondary antibody solution containing 2.5% skim milk at room temperature for 2 hr, washed with TBST for four times. Chemi Luminol assay kit, Chemilumi-one super (Nacalai Inc #02230-30, Kyoto, Japan), or Chemilumi-one ultra (Nacalai Inc #11644-24, Kyoto, Japan) was used to visualize protein bands using an ImageQuant LAS4000 imager (GE Healthcare #28955810, Chicago, IL).

For endogenous DSB-1 immunoblotting experiments, gel electrophoresis was performed using 4–12% Novex NuPage gels (Invitrogen #NP0322BOX, Waltham, MA). Proteins were transferred to a PVDF membrane at 4°C, 80 V for 2.5 hr. The membrane was blocked at room temperature for 1 hr in TBST containing 5% skim milk and then probed with primary antibody solution containing 5% skim milk at 4°C overnight or at room temperature for 2 hr, washed three times with TBST containing 1% skim milk, probed with secondary antibody solution containing 1% skim milk at room temperature for 2 hr, and washed with TBST for three times before proceeding to detection.

## Immunofluorescence and imaging

Immunostaining was performed as described in *Phillips et al., 2009*, with modifications as follows: Young adult worms (24 hr post-L4 larval stage) were dissected in 15 µL EBT (27.5 mM HEPES pH 7.4, 129.8 mM NaCl, 52.8 mM KCl, 2.2 mM EDTA, 0.55 mM EGTA, 1% Tween, 0.15% Tricane) buffer, fixed by adding another 15 µL fixative solution (25 mM HEPES pH 7.4, 118 mM NaCl, 48 mM KCl, 2 mM EDTA, 0.5 mM EGTA, 1% formaldehyde) and mixing for no more than 2 min in total on each coverslip. The excess liquid was pipetted off with 15 µL remaining which was picked up by touching a micro slide glass (Matsunami #S9901, Osaka, Japan) to the top of it before freezing at –80°C. The slides were fixed in –20°C methanol for exactly 1 min, transferred to PBST (PBS and 0.1% Tween) immediately, and washed three times (10 min/time) by moving slides to fresh PBST at room temperature. Then the slides were blocked in 0.5% BSA in PBST for 30 min. Primary antibody incubation was performed at 4°C overnight while secondary antibody incubation was performed for 2 hr at room temperature. At last each slide was mounted with 15 µL mounting medium (250 mM Tris, 1.8% NPG-glycerol) onto clean Matsunami No. 1 ½ (22 mm$^2$) coverslip.

Images were captured by a Deltavision personalDV microscope (Applied Precision/GE Healthcare, Chicago, IL) equipped with a CoolSNAP ES2 camera (photometrics) at a room temperature of 20–22°C, using a 100× UPlanSApo 1.4NA oil immersion objective (Olympus, Tokyo, Japan) and immersion oil (LaserLiquid; Cargille, Cedar Grove, NJ) at a refractive index of 1.513. The Z spacing was 0.2 µm and raw images were subjected to constrained iterative deconvolution followed by chromatic correction. Image acquisition and deconvolution was performed with the softWoRx suite (Applied Precision/GE Healthcare, Chicago, IL). Image postprocessing for publication was limited to linear intensity scaling and maximum-intensity projection using OMERO (*Burel et al., 2015*).

## FISH and quantification

The pairing on the right arm of chromosome V was monitored with FISH probes that label the 5S rDNA locus as described in *Phillips et al., 2009*, with modifications as follows: young adult worms (24 hr post-L4 larval stage) were dissected in 15 µL EBT buffer and fixed by adding another 15 µL 1% paraformaldehyde for 1–2 min. The excess liquid was removed before freezing. The slides were fixed in –20°C methanol for exactly 1 min, transferred to 2× SSCT (300 mM NaCl, 30 mM Na citrate pH 7, 0.1% Tween) immediately and washed three times (5 min/time) by moving slides to fresh 2× SSCT at room temperature. Next, the slides were put in a Coplin jar filled with EBFa (25 mM HEPES pH 7.4, 118 mM NaCl, 48 mM KCl, 2 mM EDTA, 0.5 mM EGTA, 3.7% formaldehyde) for another 5 min. After that, the slides were transferred to 2× SSCT and washed for three times (5 min/time) to remove the fixative. The slides were put into 50% formamide in 2× SSCT, incubated 10 min at 37°C, and then transferred to a new jar with the same solution, incubated at 37°C overnight. The probe solution (15 µL) was added onto a 22 × 22 mm$^2$ coverslip. The worms on the slides were touched to the drop of probe solution on the coverslip until the liquid was spreaded out. After being sealed, the slides were denatured at 95°C for 2 min 10 s and incubated at 37°C overnight. The slides were then washed with 50% formamide in 2× SSCT at 37°C twice for a total of 1 hr, and washed with 2× SSCT for 10 min, stained with DAPI, washed again with 2× SSCT, and mounted with 15 µL mounting medium onto clean Matsunami No. 1S (22 mm$^2$) coverslip. Quantification of FISH foci was done as in *Sato-Carlton et al., 2014*. FISH probes were generated as previously described (*Dernburg et al., 1998*).

## RAD-51 foci quantification

Quantitative analysis of RAD-51 foci per nucleus was performed as in *Sato-Carlton et al., 2014*. For all the genotypes except for *rad-54* and *atm-1; rad-54* mutants, manual counting was performed. For *rad-54* mutants, semi-automated counting was used as below: for early zones (1 and 2) with very few RAD-51 foci, manual counting was performed. For zones 3 and above, programs (at github.com/pmcarlton/deltavisionquant, copy archived at swh:1:rev:7faed1a32db1958b5677971c7ab-5da823d04f1c9, *Carlton, 2022*) written in GNU Octave (*Eaton et al., 2020*) were used to segment nuclei and count the number of RAD-51 foci in each nucleus. The programs proceed via the following steps: (1) nuclear centers are found by identifying all voxels in the DAPI channel whose intensity is above a threshold value (calculated with the Otsu method [*Otsu, 1975*] on a maximum-intensity projection image) that are also local maxima; these pixels are then subjected to a gravitational-type attraction that collapses clouds of pixels into small clusters that with few exceptions lie at the center

of imaged nuclei. (2) The original positions of all the pixels that contributed to a cluster that fall within a given radius of the center are used to define a three-dimensional (3D) convex hull that represents the nuclear volume. (3) Positions of all RAD-51 foci are calculated by thresholding as in step 1. (4) RAD-51 foci are assigned to the convex hull in which they are enclosed. Detected foci not located inside any convex hulls are rejected as background spots. The convex hull outlines and number of foci per nucleus are displayed as 2D projections for each image data file, and used during visual inspection of the 3D data to correct or reject mistaken counts. The nuclei on the coverslip-proximal side of the gonads were scored for each genotype. Statistical comparisons were performed via two-tailed t test.

## DAPI body counting at diakinesis

For DAPI body counting, completely resolvable contiguous DAPI positive bodies were counted in 3D stacks as described previously (*Sato-Carlton et al., 2014*). With this criterion, chromosomes that happen to be touching can occasionally be counted as a single DAPI body.

## γ-Irradiation

For DAPI body staining, late L4 larval stage worms were exposed to γ-rays for 58 min 30 s at 0.855 Gy/min (total exposure 50 Gy) in a Cs-137 Gammacell 40 Exactor (MDS Nordion, Ottawa, Canada). Irradiated worms were fixed 18–22 hr after irradiation for DAPI staining, and imaged to score DAPI-stained bodies as above. For western blotting of DSB-1, approximately 24 hr post-L4 worms were irradiated with either 10 or 100 Gy of γ-rays, and animals were lysed 1 hr post-irradiation.

## Embryonic viability scoring

To score embryonic viability and male progeny of each genotype, L4 larval stage hermaphrodites (P0s) were picked individually onto plates and transferred to fresh plates every 24 hr for 5 days. Unhatched eggs remaining on the plates 20 hr after being laid were counted as dead eggs every day. Viable F1 progeny and males were scored 4 days after P0s were removed from corresponding plates.

## AlphaFold structure prediction

Predictions were generated using the ColabFold interface (*Mirdita et al., 2021*; *Steinegger, 2022*, instantiated from github.com/sokrypton/ColabFold, commit ebf4df8) to the AlphaFold2 pipeline on the Colab platform (Google Research). Prediction was run on trimers using protein sequences for DSB-1, DSB-2, DSB-3 (*C. elegans* and *C. inopinata*) retrieved from Wormbase (*Davis et al., 2022*), and Rec114 and Mei4 (*Homo sapiens* and *Saccharomyces cerevisiae*) retrieved from Uniprot (*UniProt Consortium, 2021*). Program settings and coordinate files (in PDB text format) for the predictions are provided in *Figure 6—source data 1*.

## Multiple sequence alignment

Protein sequences in the DSB-1/2 orthology group were retrieved from the *Caenorhabditis* Genomes Project (http://caenorhabditis.org/). Due to the high diversity within this group, the list was pared down to DSB-1 orthologs of 11 species in the *elegans* group (*Figure 2—figure supplement 1*) with orthologs of DSB-2 and other proteins omitted. The protein prediction of DSB-1 for *Caenorhabditis latens* was found to be incomplete, so it was reconstructed by hand from the transcripts in Bioproject PRJNA248912, from WormBase ParaSite version 14 (*Howe et al., 2017*) using sequences from the sister species *Caenorhabditis remanei* as a guide. The sequences were then aligned using the L-INS-i setting of mafft v7.487 (*Katoh and Standley, 2013*).

## Orthology clustering and gene tree inference

We downloaded protein FASTA and GFF3 files for 39 *Caenorhabditis* species and two outgroup species (*Diploscapter coronatus* and *Diploscapter pachys*) from WormBase ParaSite (*Howe et al., 2017*) and the *Caenorhabditis* Genomes Project website (http://caenorhabditis.org/). We used AGAT v0.4.0 (*Dainat et al., 2020*) to select the longest isoform for each protein-coding gene, and clustered the filtered proteins into putatively orthologous groups using OrthoFinder v2.5.4 (*Emms and Kelly, 2019*), using an inflation value of 1.3. We identified the group containing the *C. elegans* DSB-1 and DSB-2 proteins, aligned the sequences using FSA v1.15.9 (*Bradley et al., 2009*), and inferred a gene tree using IQ-TREE v2.2.0-beta (*Nguyen et al., 2015*) under the LG substitution model with gamma

distributed rate variation among sites. We visualized the resulting gene tree using iTOL (*Letunic and Bork, 2016*) and extracted branch lengths using the ETE3 Python module (*Huerta-Cepas et al., 2016*).

Gene tree of the orthogroup containing the *C. elegans* proteins DSB-1 (CELEG.F08G5.1a) and DSB-2 (CELEG.F26H11.6) was inferred using maximum likelihood (LG substitution model with gamma distributed rate variation). The DSB-1/DSB-2 duplication event is denoted by a gray circle. Branch lengths represent the number of substitutions per site; scale is shown at left. The tree is rooted on the branch subtending the *Caenorhabditis monodelphis* and *Caenorhabditis auriculariae* sequences.

## Acknowledgements

We would like to thank Andres Canela, Minami Murai, Takaya Hashimoto, Tjebbe Boersma, Yuji Yamauchi, Hao Li, and Yoko Fujita for technical assistance. We thank the *Caenorhabditis* Genetics Center, which is funded by the National Institutes of Health National Center for Research Resources, for providing many nematode strains, and we thank WormBase.

## Additional information

### Funding

| Funder | Grant reference number | Author |
|---|---|---|
| Japan Society for the Promotion of Science | 5H04328 | Peter M Carlton |
| Japan Society for the Promotion of Science | 17K15064 | Aya Sato-Carlton |
| Howard Hughes Medical Institute | | Abby F Dernburg |
| Naito Foundation | | Aya Sato-Carlton |
| Ministry of Education, Culture, Sports, Science and Technology | | Heyun Guo |

The funders had no role in study design, data collection and interpretation, or the decision to submit the work for publication.

### Author contributions

Heyun Guo, Investigation, Writing - original draft, Writing – review and editing; Ericca L Stamper, Masa A Shimazoe, Xuan Li, Liangyu Zhang, Lewis Stevens, KC Jacky Tam, Investigation; Aya Sato-Carlton, Investigation, Writing – review and editing, Methodology; Abby F Dernburg, Conceptualization, Funding acquisition, Project administration, Supervision; Peter M Carlton, Conceptualization, Data curation, Funding acquisition, Project administration, Supervision, Writing - original draft, Writing – review and editing

### Author ORCIDs

Aya Sato-Carlton  http://orcid.org/0000-0003-0593-5238
Masa A Shimazoe  http://orcid.org/0000-0002-2018-0497
Liangyu Zhang  http://orcid.org/0000-0002-2701-0773
Lewis Stevens  http://orcid.org/0000-0002-6075-8273
Abby F Dernburg  http://orcid.org/0000-0001-8037-1079
Peter M Carlton  http://orcid.org/0000-0002-5320-6024

### Decision letter and Author response

Decision letter https://doi.org/10.7554/eLife.77956.sa1
Author response https://doi.org/10.7554/eLife.77956.sa2

## Additional files

### Supplementary files

• Supplementary file 1. Table listing strains used in this study.

• Supplementary file 2. Table listing crRNAs, repair templates, and genotyping primers for transgenes generated in this study.

• Transparent reporting form

### Data availability

All data generated or analysed during this study are included in the manuscript and supporting files; numerical Source Data files have been provided for all plots and graphs.

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
