## [Editor Report]

The connection between double-strand break (DSB) formation and chromosome pairing/synapsis during meiosis is not fully understood. In this paper, the authors show that the formation of DSBs is regulated by the DNA damage response (DDR) machinery. The paper will be of interest to the broad meiosis and DDR communities.

---

## [Decision Letter]

**Decision letter after peer review:**

Thank you for submitting your article "Phosphoregulation of DSB-1 mediates control of meiotic double-strand break activity" for consideration by *eLife*. Your article has been reviewed by 3 peer reviewers, including Federico Pelisch as the Reviewing Editor and Reviewer #1, and the evaluation has been overseen by Jessica Tyler as the Senior Editor.

Essential revisions:

1) For the phosphorylation analysis, the authors should change these qualitative observations to a more quantitative method.

2) Clarify the IR dosage selection and make it explicit in each Figure.

3) Analysis of SC assembly in pph-4.1; atl-1/nT1 and pph-4.1; dsb-1(5A) mutants.

4) Comparison of the levels and kinetics of RAD-51 foci in the dsb-1 phosphomutant series.

As an additional point, and this is entirely your choice, all reviewers felt that the AlphaFold analysis adds a mechanistic understanding of the respective roles of DSB1 and DSB-2 and could be moved to the Results section.

*Reviewer #1 (Recommendations for the authors):*

Specific points:

1. What is the rationale for going straight into DSB-1? While it is a perfectly reasonable target to investigate, an explanation for the non-expert would be beneficial.

2. For clarity, what is the length of the RNAi treatments? Along this same line, Figure S1B uses a combination of allele+RNAi. The choice of depletion/loss of function in the different figures should be explicitly explained.

3. When mentioning the DSB-1 shifts in Figure S1, the authors greatly simplify these results by stating that only in the mre-11 and rad-50 mutants DSB-1 phosphorylation is absent or significantly reduced. However, htp-3 also seems to lead to significantly reduced DSB-1 phosphorylation. The authors should change these qualitative observations to a more quantitative method (maybe phospho vs non-phospho ratio?), preferably using a detection method with a decent linear range (i.e. LiCor).

4. Related to 3, there seems to be a change in DSB-1 levels accompanying the phosphorylation changes. Is there a link between phosphorylation and stability?

5. Is the DSB-1 shift lost in the 5A mutant? In fact, an analysis of the different phospho-mutants would be informative.

6. I got confused with the different IR doses used. For example, the result in Figure 1D suggests to me that 100 Gy would be the best to analyse the effect of kinase mutants, but in Figure 1E 10 Gy is used. Could this be clarified?

7. Also in Figure 1E, it would be interesting to see the effect of the single atl-1 and atm-1 mutants on DSB-1 phosphorylation.

8. In Figure 5, the shifts do not seem equal. For example, the shift in the gfp::dsb-1; pph-4.1 seems significantly higher than the other ones. Have the authors tried phos-tag gels to get some insight into the possibility of different DSB-1 phospho-isoforms? At least this should be mentioned to make it clear that this is a complex analysis in spite of analysing through a 'simple' shift in a gel. In this Figure, the difference in total DSB-1 levels in the different conditions is very big, again pointing to the possibility that stability (linked to phosphorylation or not) plays a role.

*Reviewer #2 (Recommendations for the authors):*

My specific points are listed below:

1. Throughout the paper (Figures 1A-E, 5C, and S1B-D), the status of DSB-1 phosphorylation was assessed by the band shift in western blot analyses of DSB-1. Quantification of the fast and slow migrating DSB-1 and presenting their ratio from biological replicates will strengthen this claim and show the degree to which PPH-4.1 contributes to DSB-1 dephosphorylation.

2. In Figure 1B, please explain in the text why irradiated worms were used for this experiment. While it was shown in Figure 1D that the level of DSB-1 phosphorylation is increased upon irradiation, the context needs to be provided when it was first shown in the paper.

3. In Figures 1F-H, the authors tested how ATM-1 or ATL-1 kinases oppose PPH-4.1 activity by comparing the kinetics of RAD-51 loading in various single and double mutant series and showed that reducing the gene dose of atl-1, but not the atm-1 mutation, restores RAD-51 loading of pph-4.1 mutants. Have the authors examined the band shift of DSB-1 in single mutants of atm-1 or atl-1 on western blots? This will directly test whether ATL-1, but not ATM-1, is the major kinase responsible for DSB-1 phosphorylation.

4. In Figures 1H and 2C, the kinetics of RAD-51 appearance/disappearance differ between atl-1/nT1 and pph-4.1; atl-1/nT1 and between dsb-1(5A) and pph-4.1; dsb-1(5A). The authors explained this in the text, implicating PPH-4.1 in processing recombination intermediates. Could this also (or rather) reflect the requirement of PPH-4.1 in proper synapsis and thus the delayed meiotic progression in pph-4.1 mutants? Have the authors examined SC assembly in pph-4.1; atl-1/nT1 and pph-4.1; dsb-1(5A) mutants?

5. On page 4, in the final sentence of the 1st paragraph, the authors stated, "these results indicate that introduction of DSBs into pph-4.1 mutants also leads to increased fidelity of pairing and synapsis". However, the status of SC assembly was not examined in the current work. Examining the SC structure would be also important in interpreting the rescue results with the dsb-1(1A) in Figure 5, as it will further elucidate the role of PPH-4.1 in other meiotic processes.

6. In Figure 3D, since the dose of irradiation is important for this experiment, please indicate that 50Gy of γ-ray was used on the graph.

7. When comparing the degree of rescue shown in Figure 3C and 3D, pph-4.1; dsb-1(5A) animals seem to be more proficient in chiasma formation than pph-4.1 γ-irradiated (50 Gy). How do the DSB levels compare between dsb-1(5A) and 50 Gy irradiation? Can this be explained by the level of DSBs induced?

8. In the current flow of the paper, I feel that Figure 4 doesn't add much insight into the overall story, as it is somewhat expected (it was already known that exogenous DNA breaks rescue dsb-2 mutants). Perhaps, this could be shown earlier (with Figure 2) when dsb-1(5A) was first introduced and serve as evidence to rule out the contribution from DSB-2.

9. In Figure 5, it is interesting that mutating a subset of conserved S/T-Q sites can rescue dsb-2 mutants, but not in pph-4.1 mutants. Although representative images of RAD-51 staining were shown in Figure S4, it will be great to compare the levels and kinetics of RAD-51 foci in these dsb-1 phosphomutant series (1A, 2A, 3A vs. 5A) similarly to what's shown in Figure 4.

10. In Figure 5B, in both graphs, the labels are overlaid on top of each other, it is hard to see the data point for each genotype. In the current layout of the figure, it appears that pph-4.1 data series are included in the graph on the left, although the rescue of Him phenotype was not mentioned in the text. I would suggest separating the dataset for dsb-2 and pph-4.1 and presenting separate trends with fewer data sets on each graph so that the colors and shapes of the datapoints are clearly visible.

11. In Figure 5C, please indicate in the label that two lanes on the far right are loaded with 1.7x of the samples shown in the middle.

*Reviewer #3 (Recommendations for the authors):*

In several places, the writing, and order of presentation lack a bit of clarity, and some controls are missing.

More specifically, several points should be addressed by the authors, listed below in order of appearance (the lack of line numbers and the dense text presentation did not facilitate the review process):

– Page 1, right column, 2nd paragraph: typo: "both ATM and ATR kinases"

– 2 lines below: for clarity, specify which cases you are talking about. I guess it is "in both budding yeast and mice".

– Page 1 after "scaffold for the Spo11 core complex" the authors may cite also the Garcia et al. Nature 2021, which also proposed the "platform" formed by the RMM complex for Spo11 cutting.

– Page 2, right column end of 1st paragraph, following paragraph and Figure 1B-1C and 1D: the explanation for why γ irradiation was used in some experiments should be provided here. It is not clear at all at this moment and comes too late in the paper.

– Page 3 left column and Figures 1G, H, it would be important to show the single atm mutant (was it tested but it showed no effect, justifying the more sensitive assay in the rad-54 mutant?)

– Same column and Supplemental Figure 2C: the effect of atm on rad-54 DSB levels is extremely modest and seen only at pachytene, although significant: this should be emphasized because it is an important finding. Maybe something like " "atm-1; rad-54 germlines showed…showed a level of foci slightly exceeding that of the control…".

– Same paragraph, the explanation "Since homozygous mutation of atl-1..replication errors", justifying the use of heterozygous atl-1 mutants, should appear earlier, before "In contrast we found that heterozygous mutations…"

– Same page, right column, 1st paragraph and Figure 2B, C: this is not required, but have the authors tried to make phosphomimetic mutants of DSB-1 for the same residues? This should lead to constitutively low DSB numbers, as was seen for rec114 mutants in *S. cerevisiae*. At least, this should be discussed in the Discussion.

– Same paragraph: at this point, the authors should cite the study by Falk et al. (2010) that studied the meiotic phenotype of PP4 (pph3∆) mutants, even if the pleiotropic phenotype rendered the interpretation complicated. In that mutant, no reduction of DSB numbers was observed.

– Page 4, left column, 1st paragraph and Figure 3A, B: why is the effect of the dsb-1 (5A) mutant alone on pairing not shown? Does it also show earlier pairing, maybe because of increased DSB numbers? This is an important control that would facilitate the interpretation.

– Same page, right column 2nd paragraph and Figure 5A: the combination of mutations chosen is a bit confusing. Why was the S137-S186 combination mutant not tested, since these are the only 2 conserved positions?

– Page 5 left column, last paragraph of the results: still regarding the order of presentation of the results, the fact that a smearing of GFP-DSB-1(5A) mutant is still seen should be mentioned earlier in the results, upon the first description of the 5A mutant since indeed it shows that other residues may still be phosphorylated. Does this remaining smearing in the 5A mutant depend on ATL-1 (although it shouldn't since the phosphorylation would occur in non-consensus sites)? This needs some explanation, after reordering the results presentation.

– Page 5 right column, 2nd paragraph: the part of 3D predictions using α-fold, currently in the Discussion, adds a lot to the paper and the mechanistic understanding of the respective roles of DSB1 and DSB-2. I leave it up to the authors, but it may be also moved as a paragraph to the Results section.

– Same paragraph: the nomenclature should be homogenized for the trimers: either DSB-1:2:3 or DSB-1:DSB-2:DSB-3.

– Page 6, last paragraph of the Discussion: as mentioned earlier, the fact that DSB was not reduced in the yeast pph3∆ should be mentioned, either here, or earlier (page 3 right column).

---

## [Author Response]

Reviewer #1 (Recommendations for the authors):Specific points:1. What is the rationale for going straight into DSB-1? While it is a perfectly reasonable target to investigate, an explanation for the non-expert would be beneficial.

This is a good point and we have added “DSB-1 is absolutely required for DSBs, whereas loss of DSB-2 still allows a low level of DSB initiation” to the beginning of the Results section (line #151), to justify our initial selection of DSB-1.

2. For clarity, what is the length of the RNAi treatments? Along this same line, Figure S1B uses a combination of allele+RNAi. The choice of depletion/loss of function in the different figures should be explicitly explained.

We have listed the RNAi treatment length in the methods and Results, and explained our choice to use allele+RNAi in the Results (line #157): we needed to use a balanced (mixed) population of homozygous and heterozygous mutants to obtain a large amount of material, instead of manually picking many homozygous, so used RNAi to maximally deplete PPH-4.1 protein in the heterozygotes.

3. When mentioning the DSB-1 shifts in Figure S1, the authors greatly simplify these results by stating that only in the mre-11 and rad-50 mutants DSB-1 phosphorylation is absent or significantly reduced. However, htp-3 also seems to lead to significantly reduced DSB-1 phosphorylation. The authors should change these qualitative observations to a more quantitative method (maybe phospho vs non-phospho ratio?), preferably using a detection method with a decent linear range (i.e. LiCor).

This is not exactly what is stated in our manuscript: the experiment the reviewer refers to here is one in which we show that five mutant backgrounds that lack DSB formation (*spo-11, chk-2, htp-3, rad-50, mre-11)* also have reduced or absent DSB-1 phosphorylation; of these five, only the last two (*rad-50* and *mre-11*) do not display an increase of DSB-1 phosphorylation upon γ-irradiation since RAD-50 and MRE-11 are required for ATM/ATR activation. We have tried to make the text clearer on this point.

The affinity-purified antibody required to probe westerns for untagged DSB-1 is not readily available from the Dernburg lab at this time; we attempted to use crude serum for this purpose but it was not usable. We think that the clear presence/absence criterion for the upper band is clear enough for the strong suggestion that γ-rays can induce phosphorylation of DSB-1 in mutant backgrounds that do not make breaks; in other words, that breaks lead to DSB-1 phosphorylation.

4. Related to 3, there seems to be a change in DSB-1 levels accompanying the phosphorylation changes. Is there a link between phosphorylation and stability?

We have noticed this as well, in particular the correlation between increased phosphorylation and the lower amount of DSB-1 protein observed in *dsb-2* mutants. However, since the amount of DSB-1 is not reduced in *pph-4.1* knockdown conditions, it is not a simple correlation, thus we have refrained from further speculation on this issue.

5. Is the DSB-1 shift lost in the 5A mutant? In fact, an analysis of the different phospho-mutants would be informative.

DSB-1 is a Ser/Thr-rich protein: possessing 91 residues of mostly Ser/Thr and some Tyr in the length of total 385 amino acids. While the phospho-DSB-1 band in *dsb-1(5A)* mutants is dramatically reduced, a smearing is still present in the *dsb-1(5A)* mutant, indicating some other of the 86 Ser/Thr/Tyr sites are likely still phosphorylated at lower levels. We have made the text clearer now. We were not able to examine DSB-1 band shift in 1A, 2A or 3A mutants since the affinity-purified antibodies against endogenous DSB-1 were not available anymore from the Dernburg lab. We have chosen to focus the current manuscript mostly on the 5A allele since it completely lacks ATM/ATR consensus sites and is thus the most straightforward to interpret; we would like to address this issue in the future elsewhere.

6. I got confused with the different IR doses used. For example, the result in Figure 1D suggests to me that 100 Gy would be the best to analyse the effect of kinase mutants, but in Figure 1E 10 Gy is used. Could this be clarified?

We chose 10 Gy rather than 100 Gy for that experiment because 100 Gy of irradiation leads to high levels of embryonic inviability, a condition we wished to avoid to place the result more firmly in the realm of physiological relevance; also, since the result was already clear with 10 Gy, it was also the more technically convenient experiment to perform. We have indicated in the text that 10 Gy was sufficient for the experiment, and have indicated the dose used for all experiments in Figure 1D, E; Figure 1—figure supplement 1C, D and Figure 3D, E.

7. Also in Figure 1E, it would be interesting to see the effect of the single atl-1 and atm-1 mutants on DSB-1 phosphorylation.

We agree, but such experiments are complicated by the fact that *atl-1* knockouts have massive DNA damage in the pre-meiotic proliferative germline due to unrepaired replication errors, leading to ATM hyperactivation. Our data (not shown in the manuscript) suggests phosphorylation of DSB-1 happens in both *atm-1* single and *atl-1* single null mutants; but since the genetic data shows that *atl-1* and not *atm-1* mutation rescues *pph-4.1* mutants, combined with the fact that *atl-1* AID-depletion leads to increased breaks specifically in meiotic prophase, we conclude that ATL-1 is the more physiologically-relevant kinase inactivating DSB-1 in the wild-type situation.

8. In Figure 5, the shifts do not seem equal. For example, the shift in the gfp::dsb-1; pph-4.1 seems significantly higher than the other ones. Have the authors tried phos-tag gels to get some insight into the possibility of different DSB-1 phospho-isoforms? At least this should be mentioned to make it clear that this is a complex analysis in spite of analysing through a 'simple' shift in a gel. In this Figure, the difference in total DSB-1 levels in the different conditions is very big, again pointing to the possibility that stability (linked to phosphorylation or not) plays a role.

We agree with this observation, and think it likely that DSB-1 may be phosphorylated on non-SQ serines and/or threonines (which make up 20% of the protein). This possibility is now explicitly stated at the end of the Results section. We have tried phos-tag gels to detect GFP-DSB-1 but failed to even detect GFP-DSB-1 consistently with phos-tag gels in our hands. We have also attempted to purify tagged DSB-1 proteins from worm lysates as well as from human cell culture systems expressing worm DSB-1 to identify all the phosphorylation sites on DSB-1 by mass spec. However, the majority of tagged DSB-1 proteins end up in a detergent-resistant, insoluble fraction during the lysis and fractionation procedures, and we were not able to recover a sufficient amount of DSB-1 protein for mass spec identification of its phospho sites. We plan to address this issue in a future study.

Reviewer #2 (Recommendations for the authors):My specific points are listed below:1. Throughout the paper (Figures 1A-E, 5C, and S1B-D), the status of DSB-1 phosphorylation was assessed by the band shift in western blot analyses of DSB-1. Quantification of the fast and slow migrating DSB-1 and presenting their ratio from biological replicates will strengthen this claim and show the degree to which PPH-4.1 contributes to DSB-1 dephosphorylation.

Where possible we have now performed and appended quantification of phosphorylated versus unphosphorylated DSB-1 bands: in Figure 1A and Figure 5C. Since affinity purified antibodies probing untagged DSB-1 protein were not available anymore in the Dernburg lab, we were unable to quantify DSB-1 bands in Figure 1B, C, E or Figure 1—figure supplement 1C, D. However, since conclusions from these panels were based on presence/absence of an upper band and not a quantitative ratio, we do not believe the lack of quantitation affects our conclusions in those cases. The quantitation we were able to perform has placed those conclusions on a sounder footing, and we thank the reviewer for the suggestion.

2. In Figure 1B, please explain in the text why irradiated worms were used for this experiment. While it was shown in Figure 1D that the level of DSB-1 phosphorylation is increased upon irradiation, the context needs to be provided when it was first shown in the paper.

We agree that showing IP samples from γ-irradiated worms (Figure 1B) without explaining the effect of γ irradiation was confusing. We have replaced Figure 1B with a blot image of the λ phosphatase assay using non-irradiated, *pph-4* RNAi treated worms.

3. In Figures 1F-H, the authors tested how ATM-1 or ATL-1 kinases oppose PPH-4.1 activity by comparing the kinetics of RAD-51 loading in various single and double mutant series and showed that reducing the gene dose of atl-1, but not the atm-1 mutation, restores RAD-51 loading of pph-4.1 mutants. Have the authors examined the band shift of DSB-1 in single mutants of atm-1 or atl-1 on western blots? This will directly test whether ATL-1, but not ATM-1, is the major kinase responsible for DSB-1 phosphorylation.

This was also suggested by the reviewer #1 (specific question #7). We agree, but such experiments are complicated by the fact that *atl-1* knockouts have massive DNA damage in the proliferative germline due to unrepaired replication errors, leading to ATM hyperactivation. Our data (not shown in the manuscript) suggests phosphorylation of DSB-1 in both *atm-1* single and *atl-1* single null mutants; but since the genetic data shows that *atl-1* and not *atm-1* mutation rescues *pph-4.1* mutants, combined with the fact that *atl-1* AID-depletion leads to increased breaks specifically in meiotic prophase, we conclude that ATL-1 is the more physiologically-relevant kinase inactivating DSB-1 in the wild-type situation.

4. In Figures 1H and 2C, the kinetics of RAD-51 appearance/disappearance differ between atl-1/nT1 and pph-4.1; atl-1/nT1 and between dsb-1(5A) and pph-4.1; dsb-1(5A). The authors explained this in the text, implicating PPH-4.1 in processing recombination intermediates. Could this also (or rather) reflect the requirement of PPH-4.1 in proper synapsis and thus the delayed meiotic progression in pph-4.1 mutants? Have the authors examined SC assembly in pph-4.1; atl-1/nT1 and pph-4.1; dsb-1(5A) mutants?

Previously we reported that chromosomes are fully but non-homologously synapsed in *pph-4.1* mutants (Sato-Carlton., 2014 PLOS Gen). Other studies have shown that loss of or defects in SC assembly do not impede DSB formation or RAD-51 loading (Colaiácovo et al. 2003), (Phillips et al. 2005). In addition, non-homologous synapsis has never been shown to lead to delayed DSB formation in *C. elegans*. We therefore think it unlikely that improper synapsis in *pph-4.1* mutants *per se* delayed RAD-51 loading. However, in the 2014 work we observed cell cycle delays in young *pph-4.1* mutants, possibly deriving from hyperphosphorylation of other PPH-4.1 targets, which could account for the delays in RAD-51 appearance we see. We now mention this possibility in the main text (line #266).

To address the reviewer’s question, we have now examined SC assembly in *pph-4.1*, *pph-4.1; atl-1/nT1* and *pph-4.1; dsb-1(5A)* mutants by SYP-2 (SC central element) and HTP-3 (axial element) staining (Figure 3—figure supplement 1). All three mutants show wild-type timing and completion/integrity of SC assembly (whether between homologous or non-homologous chromosomes). More DSBs generated either in the *atl-1/nT1* or *dsb-1(5A)* background increased the number of homologously synapsed chromosomes in *pph-4* mutants. We have added these points to the Results section (line #314) and Figure 3—figure supplement 1.

5. On page 4, in the final sentence of the 1st paragraph, the authors stated, "these results indicate that introduction of DSBs into pph-4.1 mutants also leads to increased fidelity of pairing and synapsis". However, the status of SC assembly was not examined in the current work. Examining the SC structure would be also important in interpreting the rescue results with the dsb-1(1A) in Figure 5, as it will further elucidate the role of PPH-4.1 in other meiotic processes.

We agree with this point. We have now examined homologous synapsis by SYP-2 (SC central element), HTP-3 (axial element) and ZIM-3 (Chr I and IV pairing centers) staining in *pph-4.1; dsb-1(5A)* and *pph-4.1; dsb-1(1A)* mutants (Figure 3—figure supplement 1 and Figure 5—figure supplement 2). As mentioned above, *pph-4* single mutants exhibit non-homologous synapsis, but the timing of SC assembly is comparable to the wild type. Both *pph-4.1; dsb-1(5A)* and *pph-4.1; dsb-1(1A)* mutants showed normal timing of SC assembly whereas higher level of homologous synapsis is detected only in *pph-4.1; dsb-1(5A)* mutants compared to *pph-4.1; dsb-1(1A)* mutants. The *pph-4.1; dsb-1(1A)* mutants, which have fewer DSB formation than *pph-4.1; dsb-1(5A),* did not exhibit greater rescue of non-homologous synapsis. This suggests that greater levels of DSBs, found in 5A mutants, are needed to restore homologous pairing and synapsis in the *pph-4* mutant background, in which SC proteins tend to polymerize in a promiscuous manner. This information has now been added to the main text (line #314, #572), Figure 3—figure supplement 1 and Figure 5—figure supplement 2.

6. In Figure 3D, since the dose of irradiation is important for this experiment, please indicate that 50Gy of γ-ray was used on the graph.

This change has now been made.

7. When comparing the degree of rescue shown in Figure 3C and 3D, pph-4.1; dsb-1(5A) animals seem to be more proficient in chiasma formation than pph-4.1 γ-irradiated (50 Gy). How do the DSB levels compare between dsb-1(5A) and 50 Gy irradiation? Can this be explained by the level of DSBs induced?

We have qualitatively examined the levels of RAD-51 foci in γ-irradiated *pph-4.1* and *pph-4.1; dsb-1(5A)* animals and found that the 50 Gy γ-irradiated animals actually have higher focus numbers, meaning that rescue of bivalents in *pph-4.1* mutants cannot be simply understood as a function of how many breaks are made. The timing and nature of DNA breaks are different between γ-irradiation and 5A mutation: γ-irradiation generates excess levels of single as well as double strand breaks all at once (over the approximately 1-hour period of irradiation) whereas the 5A mutation generates DSBs gradually at normal kinetics. These differences may lead to different efficiency of DSB conversion to COs in the *pph-4.1* mutant background. We now point out this difference in the Results (line #346) and discuss possibilities in the Discussion (line #551).

8. In the current flow of the paper, I feel that Figure 4 doesn't add much insight into the overall story, as it is somewhat expected (it was already known that exogenous DNA breaks rescue dsb-2 mutants). Perhaps, this could be shown earlier (with Figure 2) when dsb-1(5A) was first introduced and serve as evidence to rule out the contribution from DSB-2.

While it is true that exogenous breaks from any source would be expected to rescue *dsb-2* mutants (panel C), the main points we convey in Figure 4 (panels A and B) are (1) DSB-2 is unnecessary for the 5A allele of *dsb-1* to promote a level of DSBs that are higher than wild-type, and (2) conversely, the loss of breaks in *dsb-2* mutants depends on the presence of SQ sites in DSB-1. This then sets up Figure 5 where we address the relative importance of these SQ sites in terms of their effect on *dsb-2* mutants. In panel 4C, we show for completeness that the increased breaks induced by the 5A allele suffice to rescue the viability of *dsb-2*, in a manner similar to exogenous γ-ray-induced breaks. We would prefer to keep our figures in the present order, and have attempted to clarify these main points in Figure 4. We have added another citation of the Rosu et al. paper where this γ-ray rescue of *dsb-2* was first shown. (line #375)

9. In Figure 5, it is interesting that mutating a subset of conserved S/T-Q sites can rescue dsb-2 mutants, but not in pph-4.1 mutants. Although representative images of RAD-51 staining were shown in Figure S4, it will be great to compare the levels and kinetics of RAD-51 foci in these dsb-1 phosphomutant series (1A, 2A, 3A vs. 5A) similarly to what's shown in Figure 4.

We have now compared the levels and kinetics of RAD-51 foci between *dsb-1(1A)*, *dsb-1(2A), dsb-1(3A)* and *dsb-1(5A)* mutants in Figure 5—figure supplement 1. All the mutants containing the S186A conversion exhibit an increased number of DSBs compared to wild type either in early prophase (*1A* and *3A* mutants) or throughout the meiotic prophase (*5A* mutant), and among these the *dsb-1(5A)* mutant has an extremely high number of DSBs exceeding even the wild-type level. These results suggest that a greater number of DSBs are needed to rescue *pph-4.1* mutants compared to *dsb-2* mutants. This difference is likely because PPH-4.1 has multiple, independent roles in homologous pairing, synapsis, DSB formation and cell cycle progression as we have shown previously (Sato-Carlton et al., 2014) whereas DSB-2 functions exclusively for DSB formation. Higher levels of DSBs may be needed to restore homologous pairing and synapsis in *pph-4.1* mutant background, in which SC proteins tend to polymerize in a promiscuous manner.

In general, PP4 is known to have diverse substrates, and previous studies have shown that PP4 homologs in yeast and mammals are involved in resection of DSBs and loading of RAD-51 during mitotic and meiotic cell cycles (Kim et al., 2011; Lee et al., 2010; Villoria et al., 2019; Falk et al., 2010), raising the possibility that PPH-4.1 may also contribute to timely processing of recombination intermediates downstream of DSB formation. Conversion of recombination intermediates to COs may be inefficient in *pph-4.1* mutants.

We now discuss possible explanations for the difference of embryonic viability between *pph-4.1; dsb-1(1A)* and *dsb-2;dsb-1(1A)* mutants in the Results (line #408) and Discussion (line #571) sections.

10. In Figure 5B, in both graphs, the labels are overlaid on top of each other, it is hard to see the data point for each genotype. In the current layout of the figure, it appears that pph-4.1 data series are included in the graph on the left, although the rescue of Him phenotype was not mentioned in the text. I would suggest separating the dataset for dsb-2 and pph-4.1 and presenting separate trends with fewer data sets on each graph so that the colors and shapes of the datapoints are clearly visible.

This is an excellent suggestion and we have modified Figure 5 accordingly to be visually simpler and clearer; also, since the rescue of Him phenotypes is ancillary to the rescue of viability here, we have omitted it to save space.

11. In Figure 5C, please indicate in the label that two lanes on the far right are loaded with 1.7x of the samples shown in the middle.

This has now been done.

Reviewer #3 (Recommendations for the authors):In several places, the writing, and order of presentation lack a bit of clarity, and some controls are missing.More specifically, several points should be addressed by the authors, listed below in order of appearance (the lack of line numbers and the dense text presentation did not facilitate the review process):– Page 1, right column, 2nd paragraph: typo: "both ATM and ATR kinases"

This error has now been corrected.

– 2 lines below: for clarity, specify which cases you are talking about. I guess it is "in both budding yeast and mice".

Yes, it is supposed to be “in both mice and budding yeast”, the statement has been clarified in the text now.

– Page 1 after "scaffold for the Spo11 core complex" the authors may cite also the Garcia et al. Nature 2021, which also proposed the "platform" formed by the RMM complex for Spo11 cutting.

This paper has now been cited properly as (Johnson et al. 2021).

– Page 2, right column end of 1st paragraph, following paragraph and Figure 1B-1C and 1D: the explanation for why γ irradiation was used in some experiments should be provided here. It is not clear at all at this moment and comes too late in the paper.

We have now explained in the text (line #181) that we used irradiation to activate ATM and ATR kinases when we show the western blot result in Figure 1C and D. We have also replaced Figure 1B with a blot image from non-irradiated, *pph-4.1* RNAi treated worms.

– Page 3 left column and Figures 1G, H, it would be important to show the single atm mutant (was it tested but it showed no effect, justifying the more sensitive assay in the rad-54 mutant?)

This is correct; Li and Yanowitz (2019 Genetics) have shown that RAD-51 foci are changed very little compared to WT in *atm-1* single mutants , which is why the more sensitive *rad-54* background was used in our study. In the interest of streamlining our story we chose not to pursue single *atm-1* mutants in Figure 1G, H since we saw no interaction with *pph-4.1*.

– Same column and Supplemental Figure 2C: the effect of atm on rad-54 DSB levels is extremely modest and seen only at pachytene, although significant: this should be emphasized because it is an important finding. Maybe something like " "atm-1; rad-54 germlines showed…showed a level of foci slightly exceeding that of the control…".

This is correct and we have emphasized the “slightly” increased RAD-51 foci in *atm-1;rad-54* mutant compared to control in late pachytene the text now (line #233).

– Same paragraph, the explanation "Since homozygous mutation of atl-1..replication errors", justifying the use of heterozygous atl-1 mutants, should appear earlier, before "In contrast we found that heterozygous mutations…"

This is right and we have moved this explanation up to before Figure 1G and H, where we described the effect of *atl-1* heterozygous mutation (line #211).

– Same page, right column, 1st paragraph and Figure 2B, C: this is not required, but have the authors tried to make phosphomimetic mutants of DSB-1 for the same residues? This should lead to constitutively low DSB numbers, as was seen for rec114 mutants in *S. cerevisiae*. At least, this should be discussed in the Discussion.

Yes, we have made *dsb-1* phosphomimetic mutants in which the five serines are substituted with either aspartic acid or glutamic acid. However, both of these *dsb-1* phosphomimetic mutants exhibit wild-type levels of RAD-51 foci and embryonic viability. A plausible explanation is that these phosphomimetic mutations do not functionally simulate the phosphorylation of DSB-1. We have included this information in the main text (line number #282).

– Same paragraph: at this point, the authors should cite the study by Falk et al. (2010) that studied the meiotic phenotype of PP4 (pph3∆) mutants, even if the pleiotropic phenotype rendered the interpretation complicated. In that mutant, no reduction of DSB numbers was observed.

This has been now cited.

– Page 4, left column, 1st paragraph and Figure 3A, B: why is the effect of the dsb-1 (5A) mutant alone on pairing not shown? Does it also show earlier pairing, maybe because of increased DSB numbers? This is an important control that would facilitate the interpretation.

We agree it is important to show the pairing of *dsb-1(5A)* single mutants as a control. We have now added the FISH data of *dsb-1(5A)* single mutant in Figure 3A and B, which shows wild-type timing of pairing.

– Same page, right column 2nd paragraph and Figure 5A: the combination of mutations chosen is a bit confusing. Why was the S137-S186 combination mutant not tested, since these are the only 2 conserved positions?

We generated the *dsb-1* non-phosphorylatable mutant series stepwise using CRISPR-Cas9 gene editing, and the maximum length of synthesized homology templates limited our ability to create all possible combinations. While it could be informative to test further mutations, we believe the current set suffices to support our main conclusions.

– Page 5 left column, last paragraph of the results: still regarding the order of presentation of the results, the fact that a smearing of GFP-DSB-1(5A) mutant is still seen should be mentioned earlier in the results, upon the first description of the 5A mutant since indeed it shows that other residues may still be phosphorylated. Does this remaining smearing in the 5A mutant depend on ATL-1 (although it shouldn't since the phosphorylation would occur in non-consensus sites)? This needs some explanation, after reordering the results presentation.

Thank you for the suggestion. Although we introduce the *dsb-1(5A)* mutation’s effect in Figure 2 first, we prefer to keep the western blot analysis of 5A mutants compared to controls and *dsb-2* mutants in Figure 5, because the blot image highlights the effect of each genotype on the relative ratio of phos/unphos DSB-1. Therefore we would like to keep the current flow of the story. Instead of changing the order of figures, we have made it clearer in the text that DSB-1 is a Ser/Thr-rich protein (non-SQ serines and/or threonines make up 20% of the protein) and it is very possible that DSB-1 could be phosphorylated at these non-SQ sites. (line #441)

– Page 5 right column, 2nd paragraph: the part of 3D predictions using α-fold, currently in the Discussion, adds a lot to the paper and the mechanistic understanding of the respective roles of DSB1 and DSB-2. I leave it up to the authors, but it may be also moved as a paragraph to the Results section.

We agree and we have moved it to the Results section now. Thank you for the suggestion.

– Same paragraph: the nomenclature should be homogenized for the trimers: either DSB-1:2:3 or DSB-1:DSB-2:DSB-3.

We have standardized referring to the models as (e.g.) "DSB-1:DSB-2:DSB-3".

– Page 6, last paragraph of the Discussion: as mentioned earlier, the fact that DSB was not reduced in the yeast pph3∆ should be mentioned, either here, or earlier (page 3 right column).

Thank you for pointing out this caveat. We now state that DSB levels were not reduced in *pph3∆* mutants and cite the Falk et al., 2010 paper in the first paragraph of Discussion. (line #510)